# Visualized procollagen Iα1 demonstrates the intracellular processing of propeptides

Toshiaki Tanaka[1], Koji Moriya[1], Makoto Tsunenaga[2], Takayo Yanagawa[3], Hiromi Morita[4], Takashi Minowa[4], Yoh-ichi Tagawa[1], Nobutaka Hanagata[4], Yutaka Inagaki[3], Toshiyuki Ikoma[5]

The processing of type I procollagen is essential for fibril formation; however, the steps involved remain controversial. We constructed a live cell imaging system by inserting fluorescent proteins into type I pre-procollagen α1. Based on live imaging and immunostaining, the C-propeptide is intracellularly cleaved at the perinuclear region, including the endoplasmic reticulum, and subsequently accumulates at the upside of the cell. The N-propeptide is also intracellularly cleaved, but is transported with the repeating structure domain of collagen into the extracellular region. This system makes it possible to detect relative increases and decreases in collagen secretion in a high-throughput manner by assaying fluorescence in the culture medium, and revealed that the rate-limiting step for collagen secretion occurs after the synthesis of procollagen. In the present study, we identified a defect in procollagen processing in activated hepatic stellate cells, which secrete aberrant collagen fibrils. The results obtained demonstrated the intracellular processing of type I procollagen, and revealed a link between dysfunctional processing and diseases such as hepatic fibrosis.

## Introduction

Collagen is a primary component of the ECM and accounts for ~30% of the total protein in the body (Hall, 1964; Lehninger, 1975), making it the most essential protein. Collagen α-chains twist to form a triple helical structure, and are widely distributed in tissues and organs such as skin, the basement membrane, cartilage, tendons, muscle, and blood vessel walls. 28 types of collagens have been identified to date, and most are categorized into two classes: fibrillar and non-fibrillar collagens. Collagens generally have specific amino acid sequences consisting of triplet repeat units of Gly-Xaa-Yaa. Some diseases caused by aberrant collagens have deletions and/or duplications of the triplet repeat units (Pace et al, 2001; Persikov et al, 2004), suggesting that slight aberrations in triplets affect normal folding and reduces the stability of helical structures.

Type I collagen is a representative fibrillar collagen, that accounts for more than 90% of collagen in humans. This type of collagen is composed of two identical α1 chains and one different α2 chain ([α1]2 [α2]1) encoded by COLIA1 and COLIA2, respectively, resulting in a heterotrimer. Type I collagen is involved in the formation of many human tissues, such as the dermis, cortical and cancellous bone, tendons, the corneal stroma, and most connective tissues, as the main component. The collagen protein is initially translated from mRNA as a prepropeptide, which includes a signal sequence at the N-terminus, and is transported into the ER. Procollagen, including N-propeptide (N-pp) and C-propeptide (C-pp), is modified in the ER at its lysine and proline residues with hydroxylases, which aid in trimerization, followed by glycosylation at hydroxylated lysines. Three modified procollagens twist from C-pp to the N-terminus (Nagata, 2003), forming a triple helical procollagen. The newly formed trimer is packaged by the COPII cage at the ER. However, because the generic COPII cage is generally less than 90 nm in diameter (Miller & Schekman, 2013; Malhotra & Erlmann, 2015), difficulties are associated with holding trimerized collagens of up to 400 nm in the cage (Burgeson et al, 1985). Trimerized collagens associate with the TANGO1 complex (Bard et al, 2006; Saito et al, 2009), which plays a role in the assembly of the bulky COPII cage via interactions with Sec12 and Sec16 and loads collagens into the cage at the ER exit site (Malhotra & Erlmann, 2015; Maeda et al, 2017). The collagens are then transferred into the Golgi apparatus where it undergoes final modification of its oligosaccharides. The final product is secreted out of the cells in secretory granules via plasma membrane protrusions called fibripositors (Canty et al, 2004; Canty & Kadler, 2005). Most of the previous studies reported that N-pp and C-pp are enzymatically cleaved after secretion by procollagen N- and C-propeptidases, respectively, to form collagen (Prockop & Kivirikko, 1995; Brodsky & Ramshaw, 1997; Lamandé & Bateman, 1999). However, these propeptides were also suggested to be processed intracellularly with electron microscopic and biochemical analyses (Canty et al, 2004; Humphries et al, 2008; Canty & Kadler, 2012), whereas this hypothesis

[1]School of Life Science and Technology, Tokyo Institute of Technology, 4259 Nagatsuta, Yokohama, Japan   [2]Shiseido Global Innovation Center, 1-2-11 Takashima, Yokohama, Japan   [3]School of Medicine, Tokai University, 143 Shimo-kasuya, Isehara, Japan   [4]Nanotechnology Innovation Station, National Institute for Materials Science, 1-2-1 Sengen, Tsukuba, Japan   [5]School of Materials and Chemical Technology, Tokyo Institute of Technology, Tokyo, Japan

Correspondence: ttanaka@bio.titech.ac.jp

has not been widely accepted. The processing step of type I procollagen is essential in vertebrates, but has not yet been elucidated.

Genetic defects in type I collagen have been strongly implicated in the development of many diseases including fibrosis, Ehlers-Danlos syndrome (EDS), and osteogenesis imperfecta (OI) (Kuivaniemi et al, 1991, 1997; Pace et al, 2001; Byers, 2001a). Mutations resulting in these disorders generally include single base substitutions, which alter a glycine to another amino acid in the triplet repeat unit of Gly-Xaa-Yaa (Bonadio et al, 1985), in-frame deletions that delete a single triplet unit (Hawkins et al, 1991; Lightfoot et al, 1992; Lund et al, 1996), or in-frame duplication, which increases the unit to six amino acids (Horwitz et al, 1999). Mutations that alter obligate triplet repeat units in the repeating structure domain resulted in misfolding and instability of collagen proteins (Vogel et al, 1988). Investigations on diseases associated with genetic defects in type I collagen showed that small mutations that broke the triplet repeat units in triple-helix regions led to the misfolding and instability of collagen proteins, which disrupted the fiber structure (Pace et al, 2001; Byers, 1993, 2001b; Persikov et al, 2004). Furthermore, alterations in amino acids in C-pp prevented pro-$\alpha$ assembly in the ER, and N-pp and C-pp were found to play roles in intra- and inter-chain formation (Pihlajaniemi et al, 1984; Bateman et al, 1989; Willing et al, 1990; Chessler et al, 1993; Lees & Bulleid, 1994; Oliver et al, 1996). Therefore, any modifications, such as the addition of a fluorescent protein tag onto procollagen, may interrupt the triplet repeat units of collagen sequences (Kadler et al, 2007), which has contributed to the current lack of information on their biosynthesis processes. The requirement of a tagged type I collagen resulted in the generation of variants of type I collagen $\alpha$2 (Col1$\alpha$2), which have tags, such as GFP, fused directly to the N-terminus of the repeating structure domain instead of N-pp (Lu et al, 2018; Morris et al, 2018; Omari et al, 2018; Pickard et al, 2018 Preprint). However, tagged Col1$\alpha$2 variants without N-pp have no step in N-pp processing and an exposed telopeptide sequence, which contains binding sites for fibrillogenesis that are required for its stabilization (Eyre et al, 1984; Prockop & Fertala, 1998), resulting in aberrant collagen fibrils. Mutations affecting the N-pp processing of Col1$\alpha$2 have been associated with EDS and OI (Byers et al, 1997; Malfait et al, 2013).

To elucidate the mechanisms underlying processing, transportation, secretion, and fibril formation, we developed a live imaging system for type I collagen $\alpha$1 by adding two fluorescent protein tags. The live cell imaging system demonstrated that the C-pp of procollagen is intracellularly cleaved and accumulates at the upside of the cell. N-pp is also intracellularly cleaved, but is transported together with the repeating structure domain of collagen into the extracellular region. Based on these results, we identified a defect in the processing of type I procollagen upon the activation of hepatic stellate cells (HSCs). The present results revealed a link between defects in the processing of procollagen and diseases such as hepatic fibrosis.

# Results

## Expression of type I pre-procollagen with two different tags

To clarify the mechanisms underlying collagen biosynthesis, such as processing, transportation, secretion, and fibril formation, we

inserted two fluorescent protein tags into the $\alpha$-1 chain of human type I pre-procollagen (hpreprocol1$\alpha$1). Although the addition of a fluorescent protein tag to procollagen has been precluded (Kadler et al, 2007), we examined regions in which GFP and mCherry tags were inserted without alterations to its stability, and constructed three expression vectors coding fluorescent protein-tagged hpreprocol1$\alpha$1 (Fig 1A). Constructs were designed not only to track the repeating structure domains, N-pp, and C-pp, but also to elucidate the region in the cell at which the processing of procollagen occurs by detecting alterations in fluorescence.

We initially investigated whether the tagged procollagen was correctly expressed in cells and formed trimers. The construct coding tagged pre-procollagen Gr-CC was transfected into NIH3T3 fibroblasts and its expression was detected using a confocal microscope. Many yellow particles, due to the green color of GFP merging with the red color of mCherry, were observed at the perinuclear region including the ER–Golgi (Fig 1B), revealing high levels of colocalization of the repeating structure domain with C-pp, which reflect expression of the procollagen. Immunoelectron microscopy with anti-GFP antibody confirmed GFP signals in the ER (Fig 1C). Comparison of the ER of cells expressing the tagged pre-procollagen protein with parental NIH3T3 cells showed that the ER is proper upon expression of the tagged pre-procollagen (Fig S1). To detect the transportation of the tagged repeating structure domain, we performed a collagen trafficking assay, in which cells were treated at 40°C for 3 h to arrest the transportation of synthesized proteins at the ER, and their continuous treatment at 32°C released the arrest in the presence of cycloheximide, resulted in the transportation of proteins into the Golgi (Kawaguchi et al, 2018). With the 40°C treatment of cells, we detected high levels of the colocalization of the tagged collagen with an ER marker, calreticulin (Fig 1D). With the 32°C treatment of cells for 3 h, tagged collagen colocalized with Sec16a, which is a marker of the ER exit site (Fig S2), resulting in colocalization with the cis-Golgi marker, GM130 (Figs S2 and S3). We also performed another collagen trafficking assay, in which cells were treated at 20°C for 3 h to arrest the transportation at the trans-Golgi (Roboti et al, 2013), and detected high levels of the colocalization of the tagged collagen with a trans-Golgi marker, TGN46 (Fig 1E). These results suggested that the tagged collagen passed through the ER quality control system (Copito, 1997; Araki & Nagata, 2011) and was transferred into the cis- and trans-Golgi, which was supported by data showing that the expression of the ER stress marker CHOP was not up-regulated in NIH3T3 cells stably expressing Gr-CC (Fig S4). Western blot analyses of cells expressing the tagged pre-procollagens revealed the size of the predicted proteins, which were ~200 kD (Fig 1F). This coincided with the predicted size of pre-procollagen with GFP and mCherry; however, its expression level was not estimated in Fig 1F because of the species specificity of the antibody. Collectively, these results indicated that tagged pre-procollagen was stably expressed inside cells. A BN-PAGE analysis was performed to investigate whether the tagged procollagens were incorporated into authentic collagen fibrils. The tagged procollagens were incorporated into complexes larger than trimers in cells, with a molecular size >600 kD (Fig 1G). The tagged procollagen proteins were expressed in the ER and took their proper conformation, so that they passed through the ER quality

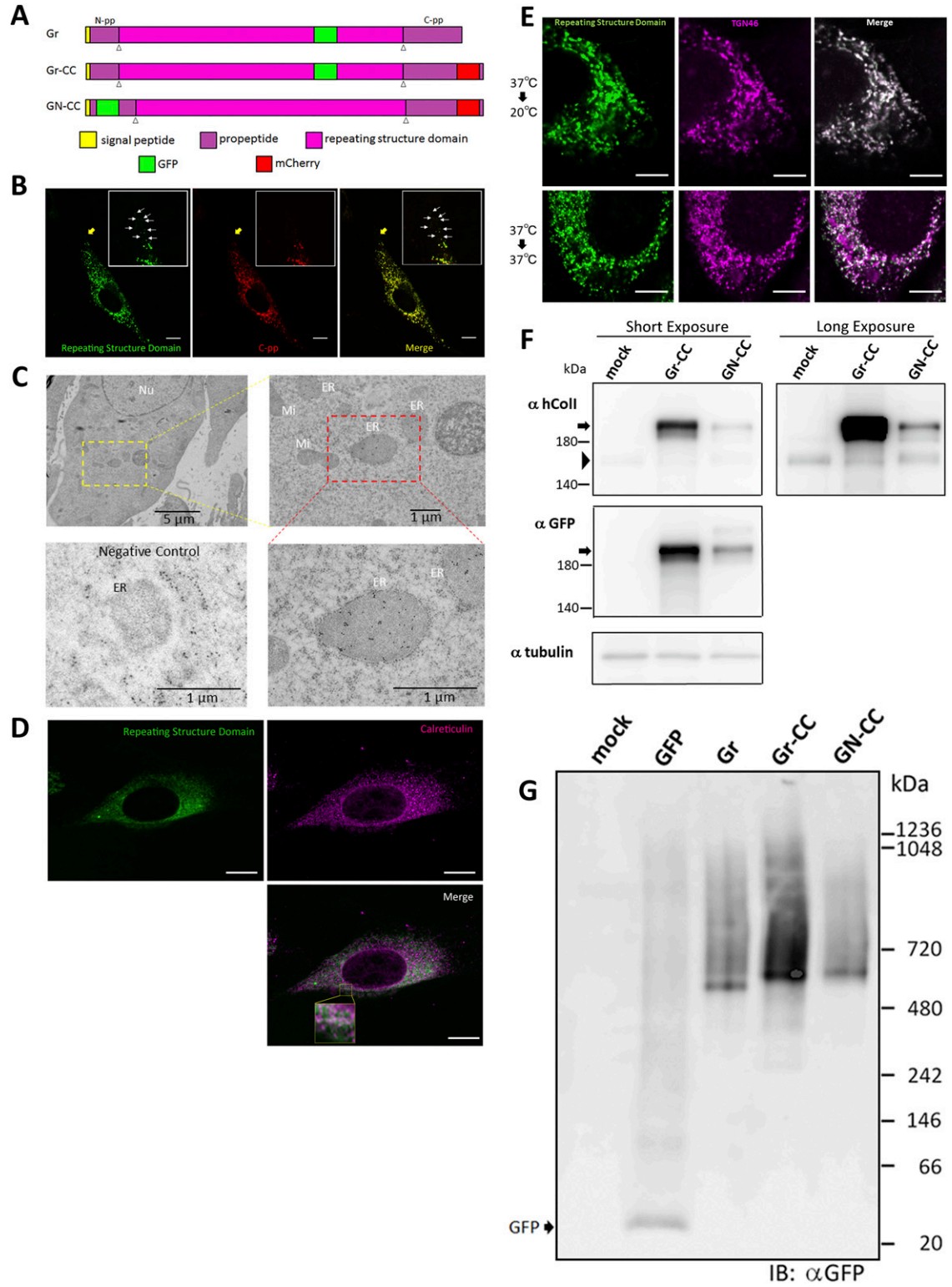

**Figure 1. Expression of the tagged type I pre-procollagen and formation of fibrils in the NIH3T3 cells.**
**(A)** Construct map for tagged type I pre-procollagen. The tagged pre-procollagen protein, Gr-CC, includes GFP in the repeating structure and mCherry in C-pp. The other constructs, including GFP in the repeating structure (designated as Gr), and GFP in N-pp and mCherry in C-pp (designated as GN-CC), were also used in this study. White triangles indicate the cleavage points. **(B)** Confocal microscopy of the tagged procollagen, Gr-CC, in a cell. Of note, the perinuclear region has procollagen in yellow and processed collagen in green indicated by arrows. Scale bars: 10 µm. **(C)** Immunoelectron microscopy of cells expressing Gr-CC. Samples were reacted with an anti-GFP antibody. The lower left is a negative control, which was reacted without the primary antibody. ER, endoplasmic reticulum; Ly, lysosome; Mi, mitochondria; Nu, nuclear. **(D)** Confocal microscopy of NIH3T3 cell expressing Gr-CC with the collagen trafficking assay. The cell was incubated at 40°C for 3 h, and immunostained with an

control system and were transferred into the cis- and trans-Golgi, resulted information of complexes larger than trimers in cells.

## Secretion of tagged type I procollagen and formation of fibrils

Fibril formation by the tagged procollagen/collagen proteins secreted into the extracellular region was examined. Confocal microscopy showed GFP signals from the repeating structure domain outside of NIH3T3 cells (Fig S5) and the pseudopodia of human fibroblast cells (Fig 2A) cultured on dishes coated with fibronectin. NIH3T3 cells with and without the tagged procollagens were then cultured for 1 mo and subjected to transmission electron microscopy (TEM). Although cells without tagged procollagen produced few fibrils, those expressing the tagged procollagens, Gr and Gr-CC, exhibited distinct fibrils between cells (Fig S6). Fibrils had a diameter of ~10 ~ 35 nm, which is reasonable for collagen fibrils (Fig 2B) (Parry et al, 1978; Kadler et al, 2007). Immunoelectron microscopy of the fibrils secreted from NIH3T3 cells, which stably expressed Gr-CC, with the anti-GFP antibody conjugated with colloidal gold particles, revealed signals on collagen fibrils with *D*-periodic structures (Kadler et al, 1996) and on narrow fibers protruding from *D*-periodic structures (Fig 2C). Atomic force microscopy (AFM) also revealed the axial repetitive structure resulting from the *D*-periodic fibrils of collagen, supporting the construction of regular collagen bundles including the tagged collagen. Therefore, the tagged type I procollagen $\alpha$1 was secreted from cells and formed *D*-periodic fibrils with an axial periodic structure specific to proper collagen fibrils.

## Intracellular processing of C-pp revealed by a live imaging analysis of cells expressing tagged procollagen

NIH3T3 cells expressing the pre-procollagen with GFP in the repeating structure domain and mCherry in C-pp (Gr-CC; Fig 1A) were subjected to time-lapse photography using a fluorescent microscope, and live cell imaging analyses were performed. Merged images of GFP and mCherry signals from cells expressing Gr-CC revealed high levels of colocalization (that appears yellow), which confirmed that the exogenous construct was translated as procollagen (Fig 3A). On live cell imaging, particles containing the processed collagen (the repeating structure domain) with the GFP tag moved from the perinuclear region including procollagen in yellow to pseudopodia, demonstrating the intracellular processing of procollagen at the perinuclear region (Video 1). Based on precise analyses by confocal microscopy, particles containing processed collagen with the GFP tag have been detected mostly in the periphery of cells, appeared in the perinuclear region, and were directed towards a pseudopodium, consistent with intracellular processing (Fig 3B and Video 2). To identify the organelle at which procollagen cleavage occurs, cells expressing Gr-CC were treated

with brefeldin A, which arrests protein trafficking at the ER, and analyzed with confocal microscopy (Fig 3C). The results obtained showed that the GFP signal was present in the ER, whereas mCherry was not, revealing the processing of procollagen at the ER. Western blot and IP-Western analyses of NIH3T3 cell lysates with the anti-GFP antibody produced signals at 150 kD, which were consistent with the processed repeating structure domain including the GFP tag, supporting the intracellular processing of propeptides (Fig S7). These results demonstrated that C-pp is intracellularly processed and the resulting repeating structure domain is secreted into the extracellular region after its transportation along pseudopodia.

## Different fates of N-pp and C-pp

Live cell imaging using cells expressing Gr-CC revealed that the most processed C-pp was not secreted (Videos 1 and 2). Therefore, we investigated the fate of C-pp after processing. Live cell imaging of the mCherry signal confirmed the accumulation of C-pp in the perinuclear region (Fig 4A and Video 3). We measured the signal intensities of mCherry and GFP in Fig 4A, and confirmed the predominant accumulation of mCherry (C-pp) in the perinuclear region and predominant secretion of GFP (the repeating structure domain) via pseudopodia (Fig S8). In a Western blot analysis of cell lysates with the anti-mCherry antibody, signals smaller than 60 kD, which corresponded to the molecular weight of C-pp+mCherry-tag, were specifically detected in the Gr-CC and GN-CC lanes, supporting the intracellular processing of C-pp (lanes 2 and 3 in Fig 4B). Processed C-pp gradually decreased in size (Fig 4B) and the medium used to culture cells included a small fragment detected by the anti-mCherry antibody (Fig 4C), suggesting that processed C-pp was degraded in cells and only a small fragment originating from processed C-pp was secreted into the extracellular region. This result was supported by data showing that a larger quantity of C-pp colocalized with lysosomes than the repeating structure domain (Fig S9).

To compare the distributions of N-pp and C-pp, cells transfected with another construct coding pre-procollagen with GFP in N-pp and mCherry in C-pp (GN-CC; Fig 1A) were used for the live cell imaging analysis. Merged images of GFP and mCherry signals from cells expressing GN-CC revealed high levels of colocalization (that appeared yellow), which confirmed the colocalization of N-pp with C-pp in the perinuclear region (Fig 4D and Video 4). On live cell imaging, particles containing processed N-pp with the GFP tag appeared in the perinuclear region in yellow and were directed towards a pseudopodium, showing that N-pp was transferred similarly to the repeating structure domain, whereas C-pp was not. This result was confirmed by measurements of the signal intensities of GFP and mCherry in Fig 4D, namely, the predominant accumulation of mCherry (C-pp) in the perinuclear region and the

---

anti-calreticulin antibody. The region surrounded by the yellow square in the lower panel was magnified. Scale bars: 10 $\mu$m (B, D). **(E)** Confocal microscopy of a NIH3T3 cell expressing Gr-CC, which was incubated at 20°C or 37°C and immunostained with anti-TGN46 antibody. Note that almost all of GFP signals were colocalized with TGN46 at 20°C in the presence of cycloheximide. Scale bars: 5 $\mu$m. **(F)** Detection of tagged procollagen by a Western blot analysis. NIH3T3 cells transfected with Gr-CC (lane 2) and GN-CC (lane 3) were lysed with NativePAGE sample buffer and subjected to analysis. Upper panel: Immunoblotted with an anti-human collagen I antibody; lower panel: Immunoblotted with the anti-GFP antibody. An identical membrane was used for reproving by another antibody to confirm identical signal positions. Arrows indicate tagged procollagens; arrowheads indicate endogenous collagen. **(G)** Detection of tagged procollagen by a blue native-PAGE analysis. Samples from (F) were subjected to the analysis and reacted with the anti-GFP antibody.

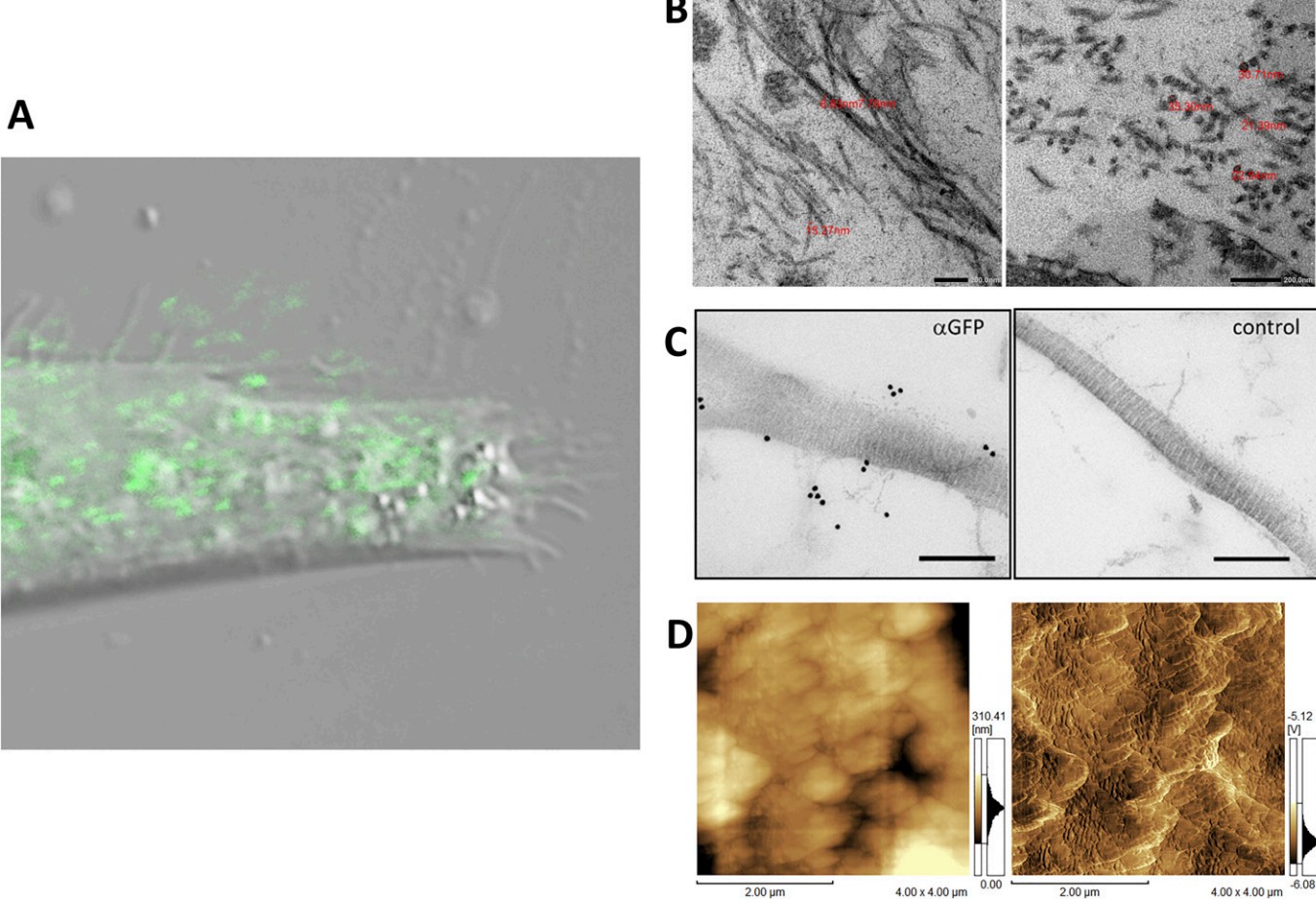

**Figure 2. Detection of extracellular fibrils of tagged type I collagen.**
**(A)** Confocal fluorescence microscopy of pseudopodia of a cell expressing the tagged collagen. Human primary fibroblast cells transfected with Gr-CC were seeded on a fibronectin-coated dish and cultured for 24 h. Magnification: ×40. Scale bar: 1 μm. Of note, the red channel to detect mCherry had no signal. **(B)** Magnification of fibrils including tagged type I collagen between cells. NIH3T3 cells transfected with Gr regulated by the CMV promoter were cultured for 1 mo and fibrils between cells were detected by transmission electron microscopy. The upper panel shows horizontal sections and the lower shows cross-sections. The red number indicates the diameter of each fibril. Magnification: ×20,000 and ×40,000, respectively. **(C)** Detection of GFP-tagged collagen secreted from NIH3T3 cells in collagen fibrils by an immunoelectron microscopic analysis. The left panel shows fibrils among cells stably expressing Gr-CC detected by an anti-GFP antibody followed by a 15-nm gold-anti-rabbit antibody. The right panel shows the negative control without the primary antibody. Scale bars: 200 nm. Note that colloidal gold signals were specifically detected in the left panel. **(D)** Detection of collagen fibrils secreted from NIH3T3 cells expressing Gr as in (B) by atomic force microscopy. Left: original image. Right: contrast was adjusted. The axial repetitive structure in the fibril reflects the collagen-specific structure (C, D).

predominant secretion of GFP (N-pp) in pseudopodia (Fig S10). Observations of full-thickness sections using confocal microscopy revealed that processed C-pp in red was distributed in the upper section of the cell (lower left panel in Fig 4D). Because N-pp and the repeating structure domain were both transported along the pseudopodium, we were unable to discriminate the region for N-pp processing from the repeating structure domain using the live cell imaging system. Therefore, we examined cells expressing GN-CC using Western blot analysis with the anti-GFP antibody. Lysates prepared from cells expressing GN-CC exhibited a specific signal at ~40 kD, which corresponded to the molecular weight of N-pp+GFP-tag, confirming that N-pp was also processed intracellularly (Fig 4E). The culture medium used for cells expressing Gr-CC produced a signal at ~150 kD with the anti-GFP antibody; however, this large fragment was not detected in the GN-CC lane (Fig 4F), supporting

the lack of N-pp in the repeating structure domain secreted into the extracellular region. The signal intensities of Western blots with lysates and the culture media of cells expressing Gr-CC detected by the anti-GFP antibody were then measured, and the ratio of collagen/procollagen was calculated. This result obtained confirmed the secretion of the repeating structure domain (Fig 4G). It is important to note that the specific signal at ~40 kD was detected in a Western blotting analysis of the culture medium of cells expressing GN-CC (Fig S11), suggesting that N-pp was secreted from cells without degradation.

## Detection of the rate-limiting step in collagen secretion

Live cell imaging by confocal microscopy was used to visualize the transportation of granules including GFP-tagged collagen along

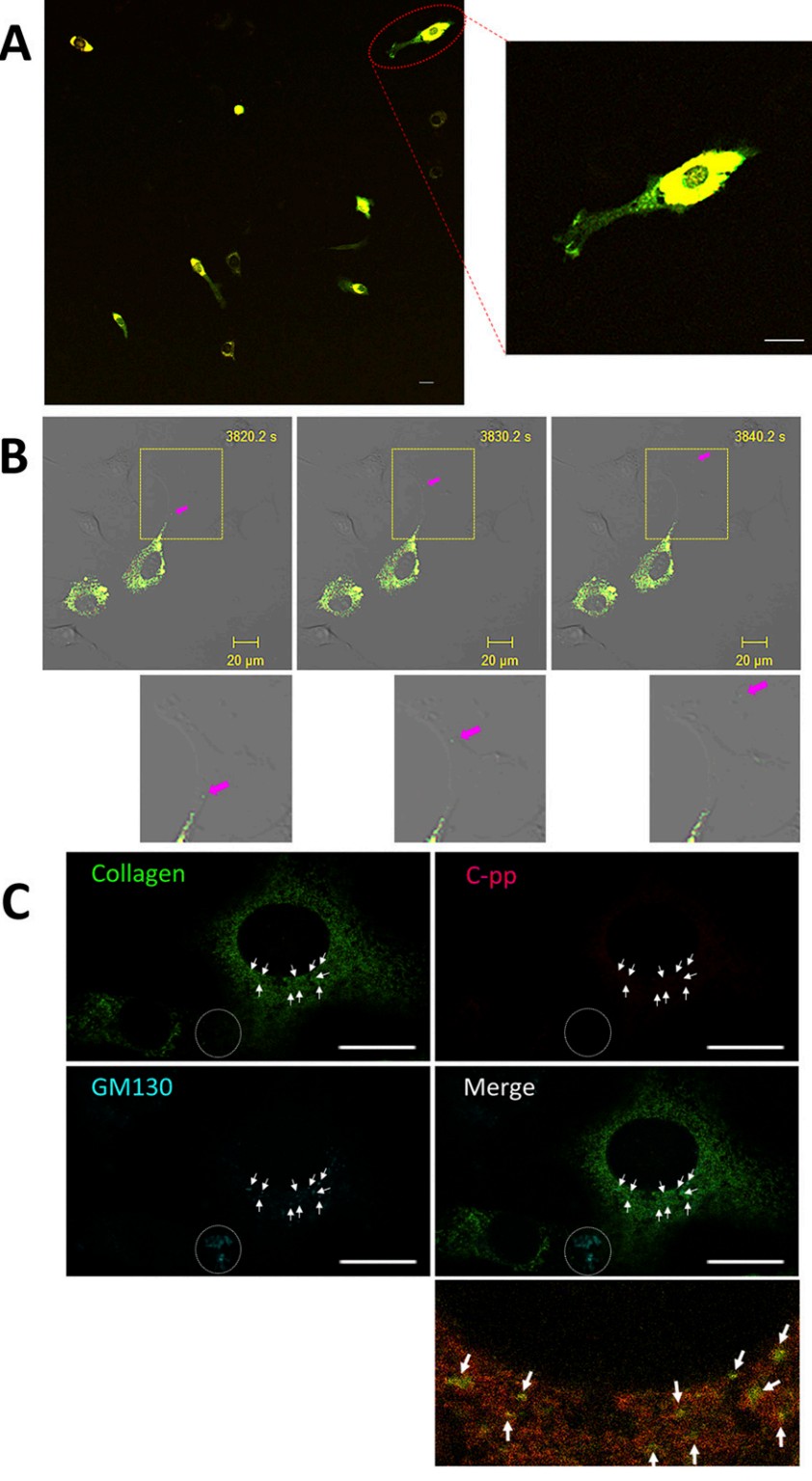

**Figure 3. Demonstration of the intracellular processing of C-pp by the live cell imaging of the tagged type I procollagen.**
**(A)** Snapshots from the live imaging of NIH3T3 cells expressing Gr-CC. The right panel shows an enlarged part of the left panel. Magnification: ×10. **(B)** Fluorescence images of NIH3T3 cells expressing Gr-CC by confocal microscopy are merged with those obtained by differential interference microscopy (DIM). Magnification: ×20. Of note, particles around the perinuclear region are both yellow and green. Green particles already lack C-pp. Each lower panel shows an enlarged image surrounded by a yellow square. Arrows show an example of a green particle that was transferred along pseudopodia and secreted into the extracellular region. Scale bars: 20 μm in (A) and (B). Original movies are shown in Videos 1 and 2. **(C)** Treatment of cells expressing Gr-CC with brefeldin A. Note that every signal of collagen was recognized in the ER, whereas that of C-pp was not. The lowest panel shows an enlargement of the portion with arrows in the merged panel. The colors of GM130 and collagen changed to yellow and red, respectively, to make their independent presence obvious. The dotted circle shows the signal of GM130 in another cell. Scale bars: 20 μm.

pseudopodia and its secretion into the extracellular region. GFP-tagged collagen was observed not only on the cell culture plate and in the intercellular spaces (Figs 2 and S5), but also in the cell culture medium (Fig 4F and Video 2). This result suggested that the

secretion of collagen from cells was detected by the GFP signal in the culture medium. Since difficulties are associated with maintaining NIH3T3 cells at full confluency, we used osteoblast-like MC3T3-E1 cells to examine the stability of the secretion of

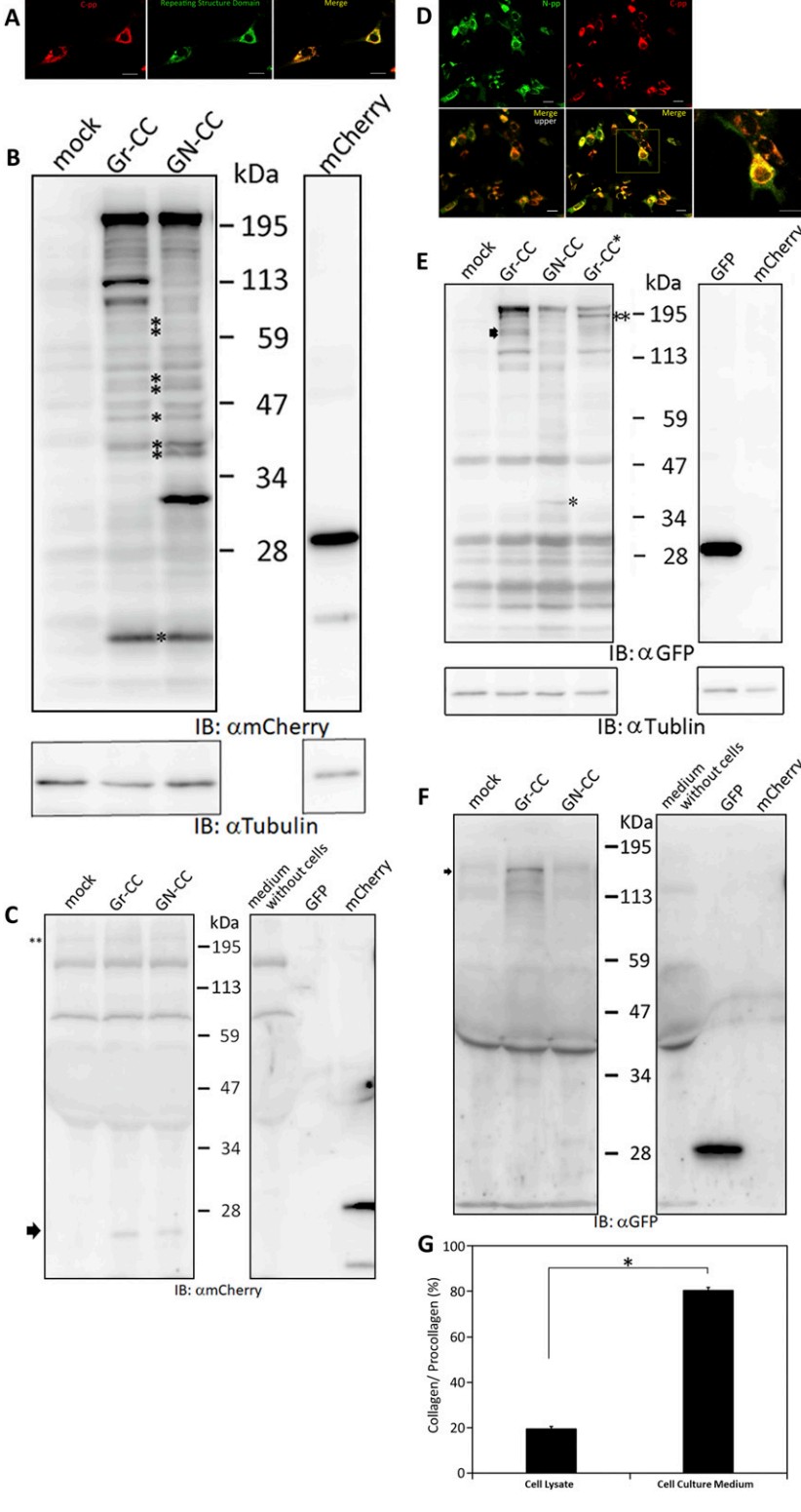

**Figure 4. Comparison of N-pp and C-pp of type I procollagen in NIH3T3 cells.**
**(A)** Snapshots indicating the intracellular distribution of C-pp (left) and the repeating structure domain (middle). The two results were merged in the right panel. ×20 magnification. The original live imaging of C-pp is shown in Video 3. The cell on the right transferred the repeating structure domain by pseudopodium, whereas C-pp accumulated in the perinuclear region. **(B)** Western blot analysis of C-pp. Lysates prepared from cells expressing Gr-CC or GN-CC were separated by SDS–PAGE and reacted with an anti-mCherry (upper panel) or anti-tubulin (lower panel) antibody, respectively. *: specific signals for C-pp+mCherry. The right panel shows the control lysate of cells expressing mCherry. **(C)** Detection of secreted proteins, including mCherry, in the culture medium. Cell culture media used for cells expressing mock, Gr-CC, and GN-CC were separated by SDS–PAGE and reacted with the anti-mCherry antibody. The right panel shows control media without cells, and the lysates of cells expressing GFP and mCherry. **(D)** Intracellular distribution of N-pp and C-pp. Snapshots from the live imaging of NIH3T3 cells expressing GN-CC. The original live imaging of the merged panel is shown in Video 4. The lower left panel shows the upper section of cells expressing GN-CC visualized on confocal microscopy. Note that the cell top is red. The lower right panel shows an enlarged part of the lower middle panel. Scale bars: 20 μm in (A) and (D). **(E)** Western blot analysis of N-pp. The same experiments were performed as in (B) with an anti-GFP antibody. In the present study, Gr-CC lacking the processing site for C-pp (designated as Gr-CC*) was used to detect the fragment consisting of the repeating structure domain+C-pp. *: a specific signal in GN-CC. **: a signal relative to the fragment consisting of the repeating structure domain+C-pp. Arrows indicate the fragments consisting of the N-pp+repeating structure domain and the repeating structure domain only. **(F)** Detection of the secreted protein including GFP in the culture medium. The same culture media as in (C) were reprobed with the anti-GFP antibody. The arrow indicates the processed repeating structure domain with the GFP tag in the Gr-CC lane. **(G)** Comparison of the collagen/procollagen ratio between the lysate and culture medium of cells expressing Gr-CC. The signal intensities of Western blot analyses with the anti-GFP antibody were measured. Each value represents the mean ± SD of triplicate measurements. *P < 0.01. All experiments were performed using NIH3T3 cells (A, B, C, D, E, F, G).

GFP-tagged collagen in long-term experiments. To obtain reproducible data, we established another cell line, Gr-CC/MC3T3, which stably expresses tagged pre-procollagen Gr-CC, and assessed the GFP signal in the culture medium. The GFP signal in the medium, in which differentiated Gr-CC/MC3T3 or parental MC3T3-E1 cells were cultured for several days, was measured using a fluorescence plate reader. Fluorescence intensity in the medium of Gr-CC/MC3T3 cells was constitutively stronger than that of parental cells for 24 d (Fig 5A).

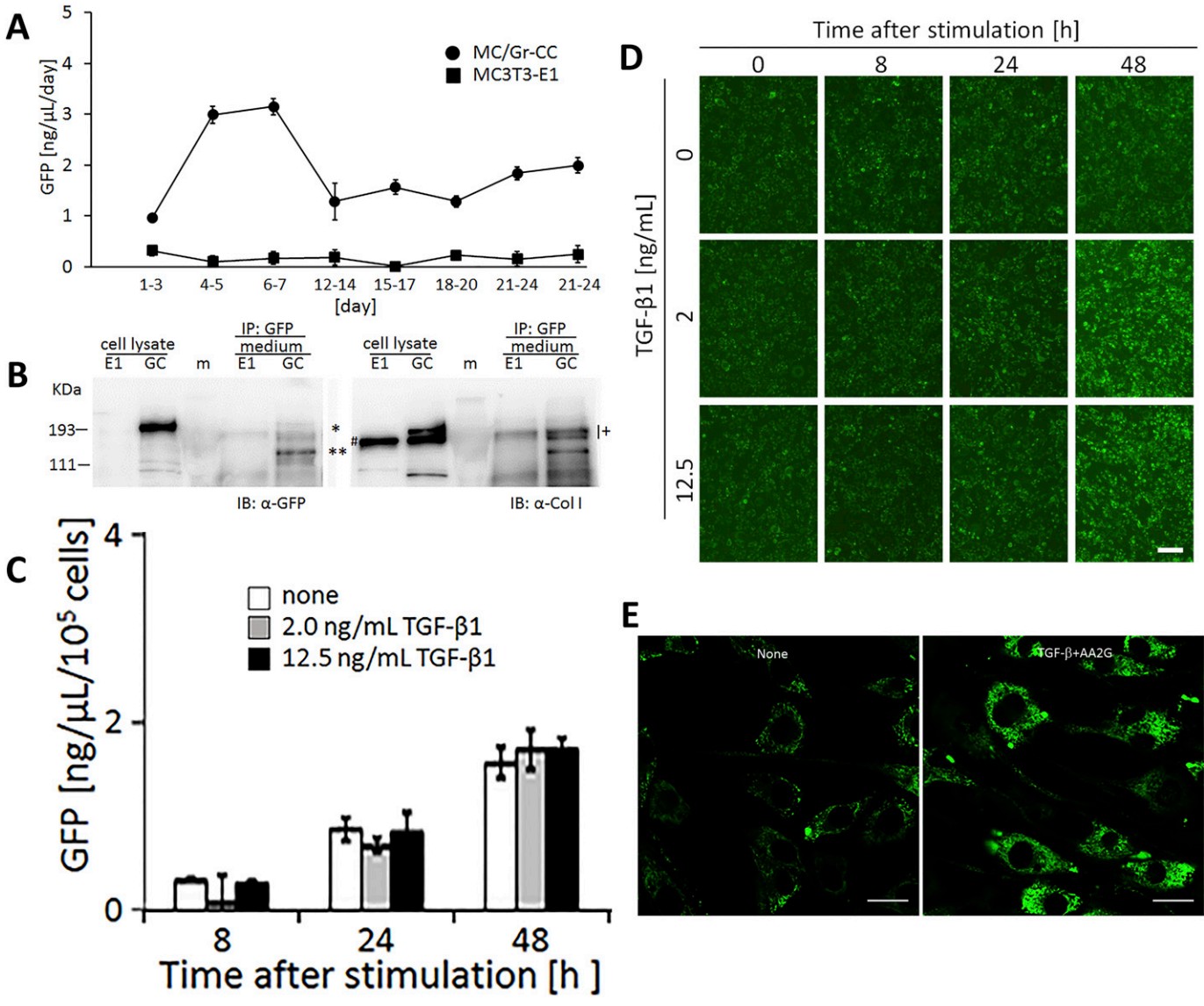

**Figure 5. Quantification of collagen secretion using tagged collagen.**
**(A)** Quantification of secreted collagen indicated by the GFP signal in the cell culture medium of Gr-CC/MC3T3. The concentration of the GFP protein was calculated using a standard curve. Each value represents the mean ± SD of triplicate measurements. **(B)** IP-Western analysis of the cell culture medium. The medium used in (A) was immunoprecipitated (IP) with an anti-GFP antibody. Immunoprecipitates were subjected to immunoblotting using the anti-GFP or anti-Col I antibody. E1: parental MC3T3-E1 cells. GC: Gr-CC/MC cells. *, procollagen with the GFP tag. **, processed repeating structure domain with the GFP tag. #, endogenous type I procollagen. +, non-specific signals. **(C)** Quantification of secreted collagen indicated by the GFP signal in the culture medium of NIH3T3 cells expressing colIp_Gr-CC. NIH3T3 cells expressing colIp_Gr-CC were stimulated with 2.0 or 12.5 ng/ml of TGF-β in the presence of AA2G. Cell numbers in each experiment are shown in Fig S15. Each value represents the mean ± SD calculated from triplicate measurements of GFP intensity in the medium of colIp_Gr-CC/NIH3T3 after subtracting the background GFP signal in the medium of parental NIH3T3 cells. **(D)** Visualization of collagen expression. Cell culture conditions are described in (C). GFP signals are shown. Magnification: ×20. Scale bar: 100 μm. **(E)** Confocal microscopy of colIp_Gr-CC/NIH3T3 cells treated with TGF-β in the presence of AA2G at 48 h. Scale bars: 20 μm. Objective lens, Zeiss Plan-APOCHROMAT 63×/1.4.

The results of the Western blot analysis confirmed that the culture medium included the repeating structure domain with GFP (Figs 5B and S12). The calibration of fluorescence intensities using purified GFP proteins in the medium demonstrated that it was possible to quantitatively measure nanogram/milliliter concentrations of tagged collagen. The concentration of the secreted tagged collagen corresponded to that in human blood (Kikuchi et al, 1994), suggesting that the secretion efficiency of tagged collagen was sufficient.

The expression of *COLI* gene is regulated in accordance with the stimulation of cells with a factor. TGF-β is a representative activator for collagen production by the transcription factor AP-1 (Jimenez et al, 1994). To establish whether the up-regulation of collagen production in cells activated by TGF-β is detected using the measurement of the GFP signal in the medium, the promoter to regulate the expression of Gr-CC was changed to that for the human *COLIA1* gene (Jimenez et al, 1994), and another stable cell line expressing the novel DNA construct was established using NIH3T3 cells. The

NIH3T3 cell line, colIp_Gr-CC/NIH3T3, was treated with TGF-β in the absence of ascorbic acid 2-glucoside (AA2G), which increases the stability of the collagen protein and is sometimes used instead of ascorbic acid, to detect the direct reaction of the promoter. Fluorescence intensity increased in TGF-β-treated cells, indicating the activation of the promoter by the TGF-β as expected (Fig S13 upper panel; Jimenez et al, 1994). The Western blot analysis of cell lysates supported increases in endogenous and tagged procollagens upon the TGF-β stimulation (Fig S13 lower panel). However, no increase in secreted collagen upon the TGF-β stimulation was noted in the Western blot analysis or measurement of fluorescence intensity in the culture medium (Fig S14). Therefore, we treated the cells with TGF-β in the presence of AA2G. Fluorescence intensity in the culture medium did not increase upon the TGF-β stimulation (Fig 5C). Although cell numbers were almost unchanged or decreased (Fig S15), the fluorescence intensity in each culture medium similarly increased according to the cultivation time (Fig 5C), reflecting the stable secretion of the tagged collagen from cells in the absence or presence of TGF-β. As shown in Fig 5D, fluorescence intensity in cells was upregulated by the TGF-β stimulation. Confocal microscopy confirmed that the synthesis of procollagen, indicated by GFP signals in the perinuclear region, was up-regulated at 48 h after the TGF-β stimulation (Fig 5E). TGF-β has been shown to up-regulate the expression of *COLIA1* gene, whereas ascorbate stabilizes collagen trimers by promoting hydroxylation at proline residues (Murad et al, 1981; Jimenez et al, 1994). Because the synthesis of procollagen increased in the perinuclear region (Fig 5E), but not in the culture medium (Fig 5C), the rate-limiting step for collagen secretion in the present study was suggested to be trimerization, processing, transportation in pseudopodia, and/or secretion from the plasma membrane.

## Defective intracellular processing of propeptides upon the activation of HSCs

To investigate defects in type I collagen in hepatic fibrosis, we introduced the Gr-CC construct into the HSC clones, CFSC-2G and -5H, which were derived from a CCl₄-induced cirrhotic rat liver (Inagaki et al, 1995). CFSC-2G cells are responsive to TGF-β and accelerate the production of type I collagen upon their activation (Inagaki et al, 1995), indicating that they are poorly activated HSC. CFSC-5H cells produce high levels of collagen and exhibit no reaction to TGF-β (Inagaki et al, 1995), which reflects the constitutive activation of these cells. A TEM analysis of collagen fibrils secreted from each cell type revealed that fibrils from CFSC-5H cells were markedly longer and thicker than those from CFSC-2G cells in the absence of TGF-β (Fig 6A), consistent with their activation levels. Live cell imaging of CFSC-5H cells expressing Gr-CC showed a different color pattern from that of NIH3T3 cells; merged images of GFP and mCherry signals from cells expressing Gr-CC revealed high levels of colocalization (appearing as yellow on Video 5), which confirmed the presence of procollagen in the perinuclear region. However, particles containing unprocessed procollagen in yellow were directed towards pseudopodia, suggesting that C-pp was not intracellularly processed (Fig 6B and Video 5). In observations of full-thickness sections by confocal microscopy, the upper section of CFSC-5H cells was dark, supporting the lack of processed C-pp in

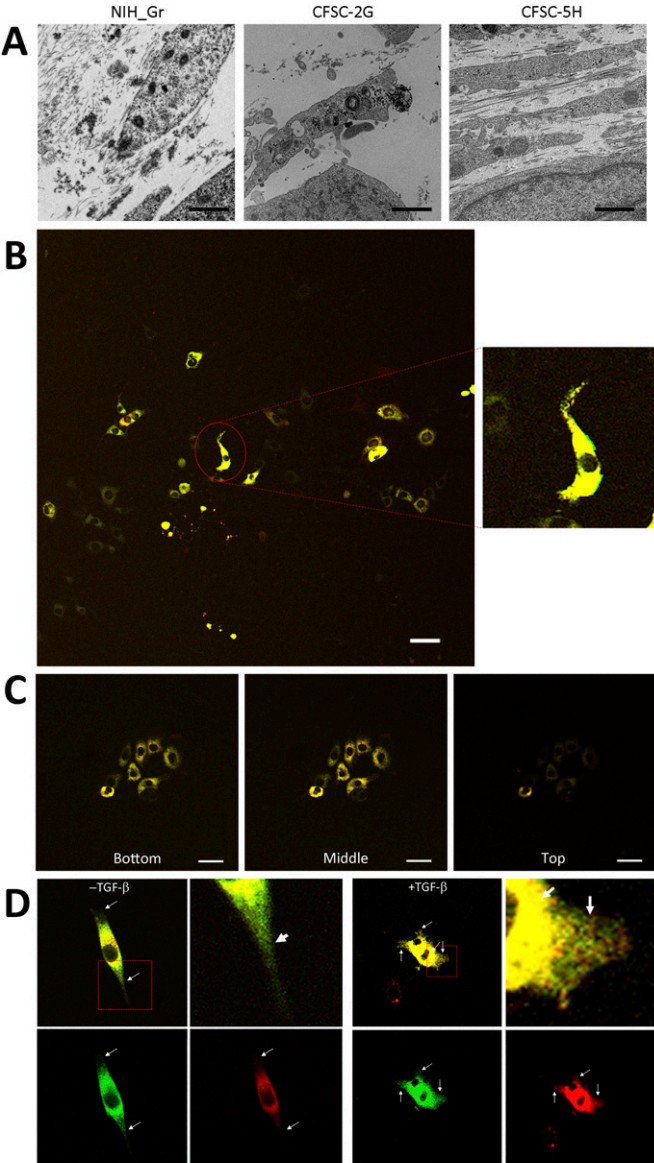

**Figure 6. Live cell imaging of tagged type I procollagen in isolated hepatic stellate cells.**

**(A)** Detection of collagen fibers by transmission electron microscopy. CFSC-2G had no or faint fibrils between cells without activation. Fibrils in CFSC-5H were longer and thicker than those in NIH3T3 cells expressing Gr. Magnification: ×5,000. Scale bars: 1.0 μm. **(B)** Snapshots from the live imaging of CFSC-5H cells expressing Gr-CC. The right panel shows an enlarged part of the left panel. Magnification: ×10. **(C)** Analysis of cells expressing tagged procollagen along the vertical axis. Comparison of colors between the bottom and top of 5H cells expressing Gr-CC. Time-lapse imaging of cells expressing tagged pre-procollagen was performed using the continuous imaging of full-thickness sections by confocal microscopy. Photos showing the top, middle, and bottom of the cell were obtained at a certain time point. Scale bars: 50 μm in (B) and (C). **(D)** Alterations in C-pp processing accompanied by the activation of stellate cells. CFSC-2G cells expressing Gr-CC were activated by TGF-β. Snapshots were obtained from the live imaging of CFSC-2G cells expressing Gr-CC in the absence (left) or presence (right) of TGF-β. Arrows show differences between them. Each upper right panel shows an enlarged part surrounded by the dotted line in the upper left panel. Yellow particles were predominant in the perinuclear region and pseudopodia of the cell treated with TGF-β. Scale bars: 10 μm. Magnification: ×10 in (B, C, D). Original movies are shown in Videos S5–S8.

cells (Fig 6C). This was also supported by the results of the Western blot analysis showing that specific signals smaller than 60 kD were not detected in the Gr-CC or GN-CC lanes with the anti-mCherry antibody (Fig S16). Furthermore, the specific signal at ~40 kD was not detected in the GN-CC lane in the Western blot analysis with the anti-GFP antibody (Fig S17), suggesting that activated HSC exhibit the defective processing of N-pp. To investigate whether defective processing in CFSC-5H cells is dependent on their activation, we examined the fluorescence signals of CFSC-2G cells expressing Gr-CC in the absence or presence of TGF-$\beta$. In the absence of TGF-$\beta$, the same color pattern as NIH3T3 cells was observed (Fig 6D left panels, Fig S18 and Video 7). However, the activation of cells by TGF-$\beta$ altered the color pattern to that exhibited by CFSC-5H cells (Fig 6D right panels, Fig S18 and Video 8). This confirmed that the activation of HSC was accompanied by a defect in the processing of propeptides.

## Discussion

The triple helical structure of collagen is constructed from C-pp to N-pp (Nagata, 2003), and these propeptides play essential roles in not only the trimerization of procollagen, but also the suppression of fibril formation during this process (Kadler et al, 1987, 2007). Therefore, the addition of a fluorescent protein as a tag to the propeptide was considered to result in its dysfunction (Lees & Bulleid, 1994; Kadler et al, 2007). Indeed, the tagging of either terminus with fluorescent proteins was previously demonstrated to result in aberrations in triple helix formation and proper secretion (Chung et al, 2009). Studies on mutations in collagen in the repeating structure domain derived from diseases, such as connective tissue disorders and OI, also revealed that a single mutation destabilizes the collagen structure and reduce its secretion (Byers, 1993; Pace et al, 2001; Persikov et al, 2004), implying that alterations in the triplet repeat unit Gly-Xaa-Yaa cause the failure of fibril formation. Therefore, the addition of a tag to collagen, particularly to principal collagen type I, has been discouraged. In the present study, we examined the positions at which GFP and mCherry tags may be introduced in the repeating structure domain and the propeptides of type I procollagen $\alpha$1 (Fig 1A). The tagged procollagen was produced in the ER (Fig 1B–D) and transferred into the cis-and trans-Golgi (Figs 1E and S2 and S3), suggesting that the tagged procollagen proteins took the appropriate conformation and were able to pass through the quality control mechanism in the ER (Copito, 1997; Araki & Nagata, 2011). We compared the ER between a parental NIH3T3 cell and that stably expressing Gr-CC, and found no marked differences (Fig S1). To establish whether a transport block of tagged collagen occurs, the ER stress marker CHOP was examined in a Western blot analysis, and the results revealed that its expression was not up-regulated in the absence of tunicamycin (Fig S4). These results support proper protein folding in NIH3T3 cells with the tagged procollagen. Procollagen in cells expressing Gr-CC assembled into multimers more than trimers (Fig 1G), and fibrils including tagged collagens formed extracellularly (Figs 2A–D and S5 and Video 2). On immunoelectron microscopy with the anti-GFP antibody, colloidal gold signals were in contact with extracellular fibril with an axial repetitive structure from D-stagger of collagen

molecules, reflecting the inclusion of GFP tags in tagged collagen in the fibrils that formed (Fig 2C). Signals on narrow fibers protruding from the D-periodic structures were also noted. Because narrow fibers were generally detected with collagen fibrils with D-periodic structures (e.g., left panel in Fig 2C), the colloidal gold signals on narrow fibers support tagged collagen being included in the proper fibril structure. Furthermore, AFM showed an axial repetitive structure in collagen fibrils with the tagged collagen (Fig 2D). These results suggest that the tagged type I procollagens assembled into proper structures with endogenous procollagens in cells, and underwent endogenous processing and secretion into the extracellular region. In the present study, intact type I procollagen $\alpha$1 was tagged. This is noteworthy because the introduction of a tagged protein instead of the N-pp of type I procollagen $\alpha$2 did not affect the secretion of tagged collagen or its fibril formation (Lu et al, 2018) even though defects in the processing of N-pp lead to EDS (Malfait et al, 2013).

The procollagen chains forming triple helical structure are generally secreted into the extracellular region, in which specific enzymes process the propeptides of N-pp and C-pp. However, the extracellular processing of procollagen has not been demonstrated, and speculation on the processing step of procollagen has been based on previous findings: procollagen is present in the cell culture medium (Goldberg, 1974; Bruns et al, 1979; Hulmes et al, 1983) and the specific enzymatic activities responsible for converting procollagen to collagen also occur in the medium (Kerwar et al, 1973; Layman & Ross, 1973). Our live cell imaging using tagged procollagen Gr-CC revealed that C-pp is processed intracellularly in the perinuclear region, including the ER and Golgi; the particles containing processed collagen with the GFP tag were present in pseudopodia and the perinuclear region, whereas those containing C-pp with the mCherry tag were predominant in the perinuclear region (Figs 1B and 3A and B and Videos 1 and 2). Cargo particles containing processed collagen with the GFP tag accumulated in the ER in brefeldin A–treated cells expressing Gr-CC, whereas those containing C-pp with the mCherry did not, showing that the intracellular processing of procollagen occurs in the ER (Fig 3C). Furthermore, Western blot analyses of the culture medium with the anti-mCherry antibody produced no specific signal at ~200 kD, which is the size of procollagen including two tags (Fig 4C), reflecting the absence of secreted procollagen with C-pp in the extracellular region. Although N-pp was also intracellularly processed (lane 4 in Fig 4E), it was transferred uniformly from the perinuclear region to the tip of pseudopodium, as observed on the live cell imaging (Fig 4D and Video 4). The specific signal at ~40 kD was detected in a Western blotting analysis of the culture medium of cells expressing GN-CC (Fig S11), suggesting the secretion of processed N-pp from cells without degradation, whereas processed C-pp was degraded in cells and only a small fragment originating from processed C-pp was secreted into the extracellular region (Fig 4C). All imaging results of tagged type I procollagen $\alpha$1 comprehensively demonstrated its intracellular processing. A previous study reported that N-pp and C-pp were processed by tolloid metalloproteases and ADAMTSs, respectively (Canty et al, 2004), which are active before the trans-Golgi network (Canty-Laird et al, 2012). The localization of processing enzymes for N-pp and C-pp is also consistent with the present results.

Although live cell imaging with tagged procollagen revealed that the particles including the repeating structure domain and N-pp unidirectionally moved from the perinuclear region to the tip of the pseudopodium (Videos 2 and 4), C-pp, which is intracellularly processed, accumulated in cells (Fig 4A and Video 3). The live cell imaging results of C-pp were consistent with those of Western blot analyses of cell lysates with the anti-mCherry antibody; specific signals at ~60 kD, which corresponded to C-pp with the mCherry tag, were detected (Fig 4B). A previous study detected the type I procollagen α1 chain tagged with GFP at C-pp at the perinuclear region including the ER and Golgi (Stephens & Pepperkok, 2002). Because the tagged procollagen also appears to be processed intracellularly, they consequently detected intracellularly processed C-pp tagged with GFP, and their findings are consistent with the observed accumulation of C-pp in the perinuclear region in the present study. However, precise analyses by confocal microscopy suggested that a small amount of C-pp fragments was secreted along the pseudopodium (Fig 4A and Video 3). Western blot analyses of the lysates of cells expressing Gr-CC/GN-CC with the anti-mCherry antibody detected specific signals smaller than 60 kD (Fig 4B), and another Western blot analysis of culture media with the anti-mCherry antibody revealed a small fragment including the antigen-recognition site that was smaller than the intact mCherry protein (Fig 4C). Therefore, a small amount of the degraded C-pp fragment was secreted from cells. This result suggests that the diagnosis based on the presence of C-pp in blood does not reflect the biosynthesis of collagen. However, the detection of N-pp is sufficient to diagnose fibrosis because N-pp was secreted together with the repeating structure domain on live cell imaging (Fig 4D and E and Video 4).

Based on live cell imaging, activated HSC, CFSC-5H cells, which secrete aberrantly thick and long fibrils of collagen (Fig 6A), exhibit defective C-pp processing (Figs 6B and S16 and Video 5). Observations of full-thickness sections of CFSC-5H cells revealed a down-regulated signal in the upper section (Fig 6C and Video 6). This result strongly supports defective C-pp processing and no accumulation of C-pp in CFSC-5H cells because processed C-pp in red was distributed in the upper section of cells (the lower left panel in Fig 4D). CFSC-2G cells processed C-pp in the absence of TGF-β (Fig 6D left panel and Video 7); however, their activation by TGF-β resulted in defective C-pp processing (Fig 6D right panel and Video 8), suggesting a strong link between defects in the processing of procollagen and fibrosis. Mutations affecting the C-pp processing of type I procollagen α1 lead to OI, which is characterized by abnormalities in connective tissue, including collagen, partly support this idea; however, the pathogenic mechanisms of OI are unknown (Lindahl et al, 2011). Although the proper removal of C-pp was previously reported to reduce the threshold concentration for fibril assembly (Kadler et al, 1987; Hulmes et al, 1989), activated HSC are considered to secrete a high concentration of collagen with C-pp that is sufficient for fibril assembly, resulting in aberrantly thick and long collagen fibrils (Fig 6A). The present results suggest that the defects observed in the intracellular processing of the propeptides of type I procollagen α1 upon cell activation resulted in the production of longer and thicker fibrils than in healthy controls, which have been implicated in many diseases such as hepatic fibrosis.

In the present study, the treatment of cells expressing Gr-CC with TGF-β increased green particles, which included the tagged repeating structure domain, in the perinuclear region (Figs 5D and E and S13); however, secreted collagen with the GFP tag did not increase in the culture medium (Figs 5C and S14), suggesting that the rate-limiting step in collagen secretion is after the synthesis of procollagen proteins. Therefore, the steps in collagen secretion, such as trimerization, processing, transportation, and secretion from the cellular membrane, are targets for the development of molecules and components to regulate collagen. Because limited information is currently available on the molecules and components that regulate these steps, many disorders that occur because of the aberration of collagen are considered to be intractable. However, the assay using collagen tagged at N-pp may not be appropriate to screen for molecules and components that regulate collagen (Wong et al, 2018) because N-pp is intracellularly processed (Fig 4E). Using our system combined with a conventional fluorescence plate reader, the secretion of collagen tagged with GFP in the repeating structure domain (Gr-CC) was sequentially quantified at nanogram/milliliter concentrations in the culture medium (Fig 5). This method enables the screening of molecules and components as possible regulators of collagen secretion in a high-throughput manner.

## Materials and Methods

### Cell culture and transient transfection

NIH3T3 cells were obtained from the RIKEN Cell Bank. Cells were cultured in DMEM supplemented with 10% fetal bovine serum, 100 units/ml of penicillin, and 100 μg/ml of streptomycin at 37°C. Sodium ascorbate or its stabilized form, AA2G, was supplied as required at concentrations of 250 μM; however, the culture medium included a sufficient amount of ascorbate to facilitate collagen biosynthesis in many cases. Cells were seeded at a density of 1.5 × 10^5 on 35-mm dishes and cultured for 24 h. Cells were treated with 1 μg of a plasmid mixed with 6 μl of polyethylenimine transfection reagent (Polyscience) following the manufacturer's instructions. Medium was changed 24 h after transfection and subjected to analysis. To detect secreted collagen, cells were cultured on fibronectin-coated dishes. The HSC clones, CFSC-2G and -5H cells, were cultured in DMEM supplemented with 10% fetal bovine serum, 1× non-essential amino acids solution (FUJIFILM Wako Pure Chemical), 100 units/ml of penicillin, and 100 μg/ml of streptomycin at 37°C. The transfection procedure was identical to that for NIH3T3 cells. Cells were stimulated with 1 ng/ml of TGF-β for 3 h as needed.

### Construction of expression plasmids coding tagged human type I pre-procollagen α1

The cDNA for human type I pre-procollagen α1 was provided by the RIKEN BioResource Center. EGFP and mCherry DNA fragments were inserted into appropriate regions in the cDNA as described in Fig 1A. We attempted more than 10 constructs, each of which had EGFP and mCherry inserted into different points of human type I pre-procollagen

**Life Science Alliance**

α1, and eventually selected the constructs in Fig 1A using the experiments described herein. Each resulting DNA fragment was inserted into the pcDNA3.1 vector. To regulate Gr-CC expression by the addition of factors, such as TGF-β, the CMV promoter was replaced with the 2.3-kbp promoter fragment of the human collagen Iα1 gene (Jimenez et al, 1994). All recombinant DNA was constructed following the guidelines of Tokyo Tech.

### Live cell imaging

A time-lapse analysis was performed using a Zeiss LSM-780 or 880 confocal microscope. The objective lens and time course are described in the text and figure legends. Fundamentally, we perform live imaging with 20× objective lens (Plan APOCHROMAT 20×/0.8, Zeiss) with large frame sizes (2,048 × 2,048 dpi), and magnify the point being observed. With an objective lens with higher magnification (63×, 100×), we lost track of the particles containing the tagged protein because of the shallow depth of the field.

### Preparation of cell extracts

48 h after transfection, cells were washed twice with PBS and lysed with EBC lysis buffer (50 mM Tris–HCl, pH 8.0, 150 mM NaCl, 1 mM EDTA, and 0.5% NP-40) supplemented with an inhibitor mix (1 ng/ml of aprotinin, 100 mM β-glycerophosphate, 1 mM NaF, 1 mM Na$_3$VO$_4$, 10 μg/ml of leupeptin, 10 μg/ml of pepstatin A, and 1 mM phenylmethylsulfonyl fluoride) or NativePAGE sample buffer (Thermo Fisher Scientific) supplemented with 10% n-dodecyl-β-D-maltoside and the inhibitor mix described above. Cell extracts were prepared as previously described (Ushio et al, 2009).

### Immunoblotting and antibodies

Immunoblotting was performed following previously described standard procedure (Li et al, 2013). Equal amounts of protein in precleared cell extracts (20–50 μg total protein) or cell culture medium were resolved by SDS–PAGE on an 8% or 10% gel after heat denaturation. Blue native polyacrylamide gel electrophoresis (BN-PAGE) was performed according to the manufacturer's protocol (Thermo Fisher Scientific). Antibodies used for immunoblotting were as follows: anti-collagen type I antibodies from Rockland and Boster Biological Technology; an anti-α-tubulin antibody (T9026) from Sigma-Aldrich; an anti-GFP antibody from Thermo Fisher Scientific; and an anti-mCherry antibody from Novus Biological. Secondary antibodies were an anti-rabbit antibody and anti-mouse Ig antibody conjugated to horseradish peroxidase from GE Healthcare. Blots were detected using SuperSignal West Femto Substrate (Thermo Fisher Scientific) and the ImageQuant LAS 4000 mini chemiluminescence detection system (GE Healthcare). In some cases, lysates prepared from NIH3T3 cells transfected with the GFP or mCherry expression vector were simultaneously separated with other samples by SDS–PAGE, and detected with the anti-GFP or mCherry antibody for confirmation.

### Immunostaining, lysosome-specific staining and ER-specific staining

Cells were fixed with MeOH and reacted with the anti-GM130 antibody (BD Transduction Laboratories) and anti-TGN46 antibody

(proteintech). In some cases, cells were cultured in the presence of 100 μg/ml cycloheximide. The procedure was previously described (Kawaguchi et al, 2018). The Cell Navigator Lysosome staining kit (Cosmo Bio) and CellLight ER-RFP, BacMam 2.0 kit (Thermo Fisher Scientific) were used to detect lysosomes and the ER in cell, respectively, and the experimental manipulation followed each manufacturer's manual. In brief, dye specifically accumulating in lysosomes or expression constructs which produce tagged ER proteins, was added to the cell-cultured medium and incubated appropriately at 37°C. After washing cells with fresh medium, cells were analyzed on a confocal microscope.

### Image manipulation and quantification of signal intensity

Image manipulation was performed with ImageJ 1.53K and Adobe Photoshop CC 2018. The adjustment of brightness and contrast with a linear algorithm was applied to the entire image. The quantification of signal intensity of an image was performed with ImageJ 1.53K.

### Electron microscopic analysis

Cells were washed twice with Hepes buffer (30 mM Hepes, 100 mM NaCl, and 2 mM CaCl$_2$), followed by fixation with 2.5% GA-HEPES buffer (2.5% glutaraldehyde in Hepes buffer) at 16°C for 2 h or at 4°C for 16 h. Fixed cells were washed six times with Hepes buffer and analyzed by AFM (JEOL Ltd.). Regarding TEM, fixed cells were stained with 1% OsO$_4$-HEPES buffer (1% osmium in Hepes buffer) for 2 h. After washing cells with 8% sucrose buffer (8% sucrose in Hepes buffer), they were serially dehydrated with 50, 70, 90, 95, and 100% ethanol. The resultant samples were embedded in epoxy resin mix (QUETOL 651: Nonenyl Succinic Anhydride = 2:1). Samples were then subjected to ultra-thin sectioning at 100 nm and observed using JEM-1400Plus (JEOL Ltd.).

### Immunoelectron microscopic analysis

Cells stably expressing Gr-CC collagen were seeded on a gold disk coated with fibronectin and cultured for 1 mo (Fig 1C). Samples were frozen in liquid propane and freeze-substituted with tannic acid in ethanol and 2% DW at −80°C for 48 h. Samples warmed to 4°C were dehydrated through anhydrous ethanol and infiltrated with LR white resin (London Resin Co., Ltd.). After incubation at 50°C for 16 h for polymerization, the samples were subjected to ultra-thin sectioning at 90 nm with a diamond knife using an Ultracut UCT microtome (Leica). Sections on nickel grids were incubated with a rabbit antibody against GFP (#ab6556; Abcam) at 20°C for 16 h. After washing with 1% BSA in PBS, they were incubated with an anti-rabbit secondary antibody conjugated with 15-nm gold particles for 2 h. Samples were washed with PBS and placed in 2% glutaraldehyde in 0.1 M phosphate buffer. After samples had dried, they were stained with 2% uranyl acetate for 5 min and subsequently placed in lead stain solution (Sigma-Aldrich Co.) at 20°C for 3 min. Samples were observed using JEM-1400Plus (JEOL Ltd.).

### Establishment of stable cell lines

MC3T3-E1 and NIH3T3 cells were cultured as described above. Cells were transfected using Lipofectamine 2000 (Thermo Fisher Scientific)

following the manufacturer's instructions. 24 h after transfection, GFP-positive cells were selected using the cell sorter SH800 (Sony Imaging Products & Solutions Inc.). Cells were seeded at a density of 3.0 × 10$^3$ on 100-mm dishes and cultured for 8 d. Cell colonies were isolated and a single cell from the colony was seeded on each well of 96-well plates. Cell clones constitutively expressing GFP were selected by a long-term culture for longer than 3 wk.

### Detection of the GFP signal in the cell culture medium

Gr-CC/MC3T3 cells were seeded at a density of 1.0 × 10$^5$ cells/well on 35-mm dishes and cultured for 24 d in the presence of 50 $\mu$g/ml of sodium ascorbic acid, 10 mM di-sodium $\beta$-glycerophosphate, 100 ng/ml of BMP-2, and 10 nM dexamethasone to induce differentiation. The culture medium was changed at the indicated time points and its GFP signal was measured using a fluorescence plate reader EnSpire (PerkinElmer). coIp_Gr-CC/NIH3T3 cells were seeded at a density of 1.0 × 10$^5$ cells/well on six-well plates and cultured for 72 h. Cells were washed twice with PBS and then cultured in fresh medium supplemented with 1% fetal bovine serum in the absence or presence of TGF-$\beta$ (2 or 12.5 ng/ml) and AA2G. The culture medium was collected at the indicated time points and precleared by centrifugation. The GFP signal in the supernatants was measured using a fluorescence plate reader. The GFP concentration in the medium was calculated using a standard curve generated by known concentrations of the GFP protein.

### Reproducibility of the data and statistical analysis

All experiments were performed more than twice with similar results and representative data are shown in the figures. The $t$ test was used to test for significance where indicated.

# Supplementary Information

# Acknowledgements

We thank Ms. Keiko Ikeda in the Biomaterials Analysis Division, Open Facility Center, Tokyo Tech., Dr. Tadashi Moro at Minophagen Pharm. Co., Dr. Yuka Madoka, and Ms. Mari Nakagawa for their experimental support. We also thank Dr. Yasuhiko Sato at Carl Zeiss Microscopy Co., Ltd. for his technical support with confocal microscopy, Drs. Kohei Kawaguchi and Masayuki Komada for providing materials, and Dr. Toshimasa Uemura for reading the manuscript. This present study was technically supported by the Open Facility Center, Tokyo Tech. and the NIMS Molecule & Material Synthesis Platform in the "Nanotechnology Platform Project" operated by the Ministry of Education, Culture, Sports, Science and Technology (MEXT), Japan (No. JPMXP09 S20NM0013). The present study was supported by the Japan Society for the Promotion of Science KAKENHI Grant numbers 26440048, 18K06013 (T Tanaka), 24241044 and 15H03526 (T Ikoma and T Tanaka), AMED Grant number JP19lm0203012, JP20lm0203012, JP21lm0203012 (T Tanaka), and Koyanagi-foundation Grant number 19060057 (T Tanaka).

## Author Contributions

T Tanaka: conceptualization, data curation, formal analysis, funding acquisition, validation, investigation, visualization, methodology, project administration, and writing—original draft, review, and editing.
K Moriya: data curation, formal analysis, validation, investigation, methodology, and writing—review and editing.
M Tsunenaga: data curation, formal analysis, validation, and investigation.
T Yanagawa: formal analysis and investigation.
H Morita: formal analysis and investigation.
T Minowa: formal analysis, validation, and investigation.
Y-i Tagawa: validation, investigation, and writing—review and editing.
N Hanagata: supervision and validation.
Y Inagaki: supervision, validation, investigation, and writing—review and editing.
T Ikoma: conceptualization, formal analysis, supervision, funding acquisition, validation, investigation, project administration, and writing—review and editing.

## Conflict of Interest Statement

The authors declare that they have no conflict of interest.

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
