## [Reviewer comments · Life Science Alliance]

Life Science Alliance

Visualized procollagen Ia1 demonstrates the intracellular processing of propeptides

Koji Moriya, Makoto Tsunenaga, Takayo Yanagawa, Hiromi Morita, Takashi Minowa, Yoh-ichi Tagawa, Nobutaka Hanagata, Yutaka Inagaki, Toshiyuki Ikoma and Toshiaki Tanaka

DOI: <https://doi.org/10.26508/lsa.202101060>

Corresponding author(s): *Dr. Toshiaki Tanaka (Tokyo Institute of Technology)*

Review Timeline:

Submission Date:	2021-02-18
Editorial Decision:	2021-07-08
Revision Received:	2021-11-22
Editorial Decision:	2021-12-23
Revision Received:	2022-01-20
Editorial Decision:	2022-01-20
Revision Received:	2022-01-26
Accepted:	2022-01-26

Scientific Editor: *Eric Sawey, PhD*

Transaction Report:

July 8, 2021

Re: Life Science Alliance manuscript #LSA-2021-01060

Dear Dr. Tanaka,

Thank you for submitting your manuscript entitled "Visualized procollagen Ia1 demonstrates intracellular processing of propeptides" to Life Science Alliance. The manuscript was assessed by expert reviewers, whose comments are appended to this letter. We invite you to submit a revised manuscript addressing the Reviewer comments.

Thank you for this interesting contribution to Life Science Alliance. We are looking forward to receiving your revised manuscript.

Sincerely,

B. MANUSCRIPT ORGANIZATION AND FORMATTING:

Reviewer #1 (Comments to the Authors (Required)):

The manuscript by Tanaka et al. uses microscopy and biochemical approaches to identify the cellular location where procollagen processing takes place in cells, and the data presented points out that the processing takes place intracellularly. There are defects in this process in disease conditions. However, there are areas where the quality of data presented is flawed and needs to be improved upon substantially to validate the conclusions. Therefore, the paper can only be accepted with the significant revisions mentioned below. If the data quality is enhanced and the claims are substantiated, the story presented here can have exciting implications in the context of collagen processing in normal and disease conditions.

1) I would recommend investing some time to present data logically and simply and make extensive text edits (some examples mentioned later), and improve the figures to make the manuscript easier to follow for the readers.

2) The organelle where the intracellular cleavage of procollagen is taking place is not completely clear. The data presented is mainly due to saturated images showing colocalization of GFP and mcherry signals in the perinuclear region. Because of that the colocalization data with the Golgi marker is not very convincing. Authors can address this question by doing three-color imaging with GM130, m-cherry, and GFP or using any other Golgi marker preferable a luminal protein. Authors have presented immune EM data where the GFP signal was shown to be in the ER. Authors should image the Golgi apparatus from those same blocks and show images of gold particles in the Golgi. Alternatively, the authors should do a 20°C block used to accumulate the cargo at the Golgi and do three-color imaging as mentioned above with a suitable Golgi marker. Finally, authors should explain the discrepancy between images 1b and 1e, which show two different cells expressing the Gr-CC construct. While the GFP (Repetitive domain) and the mcherry (Cpp) signals appear fully colocalized in Fig 1b, in Fig 1e, the mcherry signal is very faint. It seems to be distinct from the GFP signal. The mcherry (Cpp) also shows colocalization with lysosomal marker (information on which lysosomal marker was used is missing). It should be mentioned and discussed in the text.

3) The western blot (WB) with the anti-Collagen antibody is of poor quality. Even in the cells with only GFP transfection shows a faint band at a similar molecular weight and relatively same intensity with that of GN-CC. The authors should include blots with a better expression of Gr-CC and GN-CC constructs.

4) The movie S2 is nice and provides the main result that only the processed collagen is secreted upon cleavage. However, a few mcherry spots are also seen to be secreted. The authors should solidify this result by performing a WB from the secreted medium and cell lysates and probe with GFP and mcherry antibodies, respectively. Secretion assays should be ideally performed in a serum-free medium to avoid the BSA contamination, which affects WBs presented in Fig 4. As the authors further state that a small portion of Cpp gets processed and secreted, they should probe the same lane of cell lysates and supernatant with GFP and mcherry antibodies one after another and then express the data as a ratio of GFP/mcherry in cell lysates and supernatant. The expectation would be that this ratio would be much higher in the supernatant compared to the cell lysate.

5) The movie depicting the separation of Npp from Cpp is not very clear. Therefore, the authors should use the WB mentioned above strategy from cell lysates and secreted culture medium of cells expressing GN-CC construct and probing the blots using GFP and mcherry antibodies sequentially.

6) Please add a better description of the relationship between TGF β and collagen processing and secretion. Currently, it appears out of the blue in the narrative. Similarly, explain AA2G in the text, and please expand the abbreviation before the first usage.

7) The way the data is presented, I cannot see any differences in cells in Fig 6D or the supplementary movies s7 and s8 with and without TGF β . Please show higher magnification images, and please use arrows to improve the presentation to point out what differences one expected to see.

8) It must be re-emphasized that extensive edits in the text are necessary and should be scrutinized before resubmission. Some suggestions are as follows:

I would recommend to avoid expressions as on Page 19/ line 15: "region was yellow" and replace to "merged images of GFP and mCherry signal from cells expressing Gr-CC revealed high levels of colocalizations (that appears yellow on Video 1) which testifies that..." and Page 19/ line 21: "The green particles moved uni-directionally from the yellow perinuclear region to the tip of the pseudopodium..." could be changed to "vesicles containing processed procollagen with GFP tag (repeating domain structure) have been seen mostly in the periphery of the cell and were appearing in the perinuclear region and directing towards pseudopodium ". Page 23/ line 8. Page 35/line 4 and so on.

9) Scale bars are missing. Figure 1C. 1D. 6B. 6C. 6D.

Reviewer #2 (Comments to the Authors (Required)):

Collagens are necessary for the assembly of extracellular matrix and compose about 25% of our dry protein weight. The mechanism of secretion of these bulky cargoes is only recently beginning to emerge and the tagged collagens presented in this paper could be of tremendous value to the field. Protein TANGO1 has emerged as a key player in the collagen export pathway by its binding to collagens via Hsp47 and creating a novel route for their export.

This is an interesting paper and the authors are to be credited for address this important issue. The authors are encouraged to address the following issues to strengthen their conclusions.

1. They state that the newly formed trimer is packaged by the COPII cage with mono-ubiquitylated Sec31, and then transferred into the Golgi apparatus (6), where it undergoes final modification of its oligosaccharides.

The authors are encouraged to read up the literature on collagen export from the ER for accuracy and for a fair representation of the field.

2. Figure 1b. What are the green dots ? They appear to be ER exit sites? Show co-localization with an ER site marker. Use TANGO1 or Sec16.

3. Figure 1d. Very little co-localization of expressed collagens with GM130. The rest and the predominant material is probably still in the ER or ER exit sites. Please check. I would suggest the authors generate stable lines to avoid issues due to non-uniform expression of collagens.

4. Figure 3B. What is the arrow suppose to show? I do not see anything in the images shown. Is this green spot a transport carrier? Please provide better pictures at higher magnification. Provide some sort of quantitation? Are the presumed spots carriers between ER and the Golgi or post Golgi?

5. Where is the N-pp degraded after cleavage?

6. the authors should show whether the cleavage takes places in the ER or the Golgi. use of specific means to inhibit collagen egress from these compartments will help address this important concern.

Reviewer #3 (Comments to the Authors (Required)):

Tanaka et al. describe the cloning and characterisation of a human COL1A1 cDNA construct that has GFP integrated in the repeating structure domain. This work attempts to overcome the problem that N- or C- terminal tagging of COL1 does not allow monitoring the deposition of collagen in the ECM, as both, N- and C- terminus of COL1 are cleaved by proteases for proper processing of procollagen into collagen fibers.

Tagging COL1 in its repeated structure domain has remained a challenge, mainly due to concerns of proper folding and hindrance in fibril formation, which the authors of the current manuscript have summarised very well.

In the current manuscript, light as well as electron microscopy techniques have been used to analyse the cellular localisation of Gr-CC encoded COL1A1 in NIH3T3 cells. Western blot analysis was used to test the folding of Gr-CC encoded COL1A1 in cell lysates followed by measurements of GFP levels in cells, supernatants as well as ECM. An alternative construct driving the COL1A1 expression by a CMV promotor to one which should respond to TGFβ1 stimulation is also described.

If the current manuscript could convincingly show that the COL1A1 with a GFP tag in the repeating structure domain can fold properly and assemble into fibrils in the ECM, like wildtype COL1A1, it would represent an important major methodological advancement opening a number of new experimental approaches with relevance to e.g. ECM formation and regulation or development of fibrosis.

While the methods of choice to validate the claims in the current work appear suitable, a substantial fraction of the data shown are in this reviewer's view not of sufficient quality to substantiate the claims made by the authors. The following points need to be seriously addressed before this manuscript should be published:

1) Authors should publish the sequence of the insertion of the tag, this should subsequently help the community to endogenously tag COL1A1 repeating structure domain. Authors should mention why this site was chosen, was it a random choice or there was some methodology involved.

- 2) Figure 1C: A control EM micrograph showing cells mock transfected is missing. In the micrographs shown, the ER appears bloated, which suggests a transport block of the GFP tagged COL1, possibly caused by the tagging? and without a control reference, it is not possible to judge the validity of the experiment.
- 3) Figure 1D and related supplementary figure: the colocalization of Gr-CC encoded COL1A1 with GM130 is not convincing and presented images should be improved and accompanied with a quantitative evaluation of the co-localisation, e.g. by plot profiles of the two signals (GM130 and Gr-CC GFP).
- 4) In all the images throughout the manuscript, Gr-CC, the GFP and mCherry colocalise remarkably but not in figure 1e. Only one cell is shown and corresponding signals do not overlap. Again, colocalization quantification should be done. Also, co-localisation of Gr-CC and lysotracker could be done to exclude Gr-CC localising to lysosomes. Alternatively, lysosomal inhibitors such as chloroquine or bafilomycin should be used. Furthermore, levels of ER stress markers such as BiP and CHOP could be measured to ensure that indeed proper folding has occurred and that no unfolded protein response has been triggered.
- 5) Figure 1f: This reviewer is not convinced that there is a band in the Gr-CC lane upon hCol1 immuno-blot. Please show where the endogenous collagen runs, using anti-mouse collagen antibody for a reference and estimation of expression levels.
- 6) Figure 2 and Figure S2: to convincingly show that Gr-CC is deposited in the ECM, the authors should use macromolecular crowding protocol (<https://doi.org/10.3389/fmed.2020.615774>).
- 7) The movies and data showing the movement of different signals are not optimal. Most of the images are saturated. Removing the transmitted light channel and just showing individual color channels with a black background could improve the visualisation.
- 8) Figure 4a and Movie S5: both, GFP and mCherry show the same movement. A difference between N-pp and C-pp is not clear to this reviewer. Based on the data show, the claim that N-pp and C-pp have different fates is not valid.
- 9) Figure 4C: loading controls of the western blot cell lysates are missing.
- 10) Exchanging the normal CMV promotor for one which responds to TGF β is a very nice idea. Here again, the authors should submit the sequences they have used. This construct can help also accelerate the deposition of Gr-CC in macromolecular crowding experiment.
- 11) Figure 5C: data shown suggest that there is no difference in GFP levels in control and TGF β 1 treated samples and suggests to this reviewer that the data do not support the hypothesis that Gr-CC is secreted into the extracellular space. The existence of GFP in the culture supernatant (figure 5a and 5b) could also be explained to occur due to lysis of GFP positive cells over time in culture. The increase in the intracellular level in figure 5d is not reflected in Figure 5C. Therefore, based on these data, the claim that collagen has differential kinetics under disease conditons such as in hepatic fibrosis cannot be made. This data suggest more that the Gr-CC is not secreted.
- 12) Images showing the expression of Gr-CC in hepatic stellate cells are saturated.

Reviewer #1 (Comments to the Authors (Required)):

The manuscript by Tanaka et al. uses microscopy and biochemical approaches to identify the cellular location where procollagen processing takes place in cells, and the data presented points out that the processing takes place intracellularly. There are defects in this process in disease conditions. However, there are areas where the quality of data presented is flawed and needs to be improved upon substantially to validate the conclusions. Therefore, the paper can only be accepted with the significant revisions mentioned below. If the data quality is enhanced and the claims are substantiated, the story presented here can have exciting implications in the context of collagen processing in normal and disease conditions.

1) I would recommend investing some time to present data logically and simply and make extensive text edits (some examples mentioned later), and improve the figures to make the manuscript easier to follow for the readers.

--- We extensively improved the text with many figure replacements to make the manuscript easier to understand for readers. Regarding Fig 1, in which some differences among images were mentioned, the image in Fig 1b was replaced to avoid any misunderstanding regarding the distribution of GFP (the repeating structure domain) and mCherry (C-pp), and previous Fig. 1e was removed from the text because it may complicate the logic of the story. Instead of Fig 1e, we compared the ER in parental NIH3T3 cells and NIH3T3 cells expressing Gr-CC (Fig S1) and showed that the expression of CHOP was not up-regulated (Fig S4), thereby demonstrating the absence of an ER-stress response with the expression of the tagged procollagen. We also performed a collagen trafficking assay, in which the treatment of cells with 40°C followed by 32°C was conducted to artificially regulate collagen trafficking (ref. BBRC 499, 635-641, 2018), and added new Figures (new Figs 1d-f) to clearly show the movement of the tagged collagen for readers. We also revised Western blotting data in Fig. 1f by using another anti-collagen I antibody to clarify the signals. The signal intensities of images were measured and the results obtained were summarized in the graphs (Figs 4g, S8 and S10). In accordance with the Reviewer's suggestion, the manuscript was appropriately revised with the addition and deletion of data and text as below.

2) The organelle where the intracellular cleavage of procollagen is taking place is not completely clear.

--- To identify the organelle at which the intracellular cleavage of procollagen occurs, we performed an experiment in which collagen trafficking was inhibited at the ER in cells treated with bafilomycin (Fig 3c). The result obtained revealed the presence of GFP-collagen in the ER, but not mCherry-C-pp, indicating that the C-pp is cleaved in the ER. This result was explained in the revised text (page 12, line 14-18).

The data presented is mainly due to saturated images showing colocalization of GFP and mcherry signals in the perinuclear region.

--- Since signal intensity in the perinuclear region was markedly stronger than that in pseudopodia, it was difficult to show signals in pseudopodia without the overexposure of signals in the perinuclear region. Signal intensities in Figs 3a, 4a, 4d, 6b and 6d (and movies related to these figures) were adjusted to show the colors of the particles in pseudopodia; therefore, signals in the perinuclear region were slightly saturated. Image cropping around pseudopodia may be used to resolve this issue in these figures; however, since we would like to show the color alteration from yellow in the perinuclear region to green in pseudopodia, we allowed some overexposure of signals in the perinuclear region. We need to adjust the whole signal intensity in accordance with the portion to be shown with the figure, and attempted to avoid saturated images as much as possible.

To show an unsaturated image of the colocalization of GFP with mCherry in the perinuclear region as a reference, we prepared a new image in Fig 1b, in which we adjusted the signal intensity to the merged signals in yellow in the perinuclear region, which resulted in faint signals of green particles in pseudopodia (new Fig 1b, indicated by arrows). As a reference, images of hepatic stellate cells, in which GFP and mCherry signals were not saturated in the perinuclear region, are shown in revised Fig S18.

Because of that the colocalization data with the Golgi marker is not very convincing. Authors can address this question by doing three-color imaging with GM130, m-cherry, and GFP or using any other Golgi marker preferable a luminal protein.

--- We performed a collagen trafficking assay in which cells were treated at 40°C followed by 32°C to accumulate cargo at the ER and Golgi, respectively, in accordance with the suggestion by the Reviewer. We provided a better merged-image of GFP with GM130, in which the merged portion is shown as white color (new Fig 1e). Previous Fig. 1d was moved to Supplementary material (Fig S3) to show a magnified image of the merged portion. Three-color imaging with GM130, mCherry, and GFP did not help to show the colocalization of collagen with GM130, because C-pp was cleaved from collagen at the ER as shown in Fig 3c.

Authors have presented immune EM data where the GFP signal was shown to be in the ER. Authors should image the Golgi apparatus from those same blocks and show images of gold particles in the Golgi. Alternatively, the authors should do a 20{degree sign}C block used to accumulate the cargo at the Golgi and do three-color imaging as mentioned above with a suitable Golgi marker.

--- We were unable to detect the Golgi apparatus from the block of immune EM. However, as suggested by the Reviewer, we successfully showed the colocalization of GFP-tagged collagen at the Golgi using the collagen trafficking assay as explained above. We added these data as new Fig 1e to the revised manuscript.

Finally, authors should explain the discrepancy between images 1b and 1e, which show two different cells expressing the Gr-CC construct. While the GFP (Repetitive domain) and the mcherry (Cpp) signals appear fully colocalized in Fig1b, in Fig1e, the mcherry signal is very faint. It seems to be distinct from the GFP signal.

--- We apologize for presenting an inadequate image as previous Fig 1e, in which C-pp signals were not clear because of the low resolution of the image. The revised image shown as Fig S9 confirmed that GFP and mCherry colocalized in the perinuclear region.

The mcherry (Cpp) also shows colocalization with lysosomal marker (information on which lysosomal marker was used is missing). It should be mentioned and discussed in the text.

--- In accordance with the suggestion by the Reviewer, we described the colocalization of C-pp with lysosome on p14, line 2-4.

We apologize for the poor description of the staining method of lysosomes. We explained in the figure caption that lysosomes were stained with a lysosome marker (previous Fig 1e, new Fig S 9), but used the Cell Navigator Lysosome staining kit. Since the kit is generally used to show lysosomes and the dye used for the kit (LysoBriteTM) is explained in the manufacturer's manual, we added the kit name to the Materials and Methods.

3)The western blot (WB) with the anti-Collagen antibody is of poor quality. Even in the cells with only GFP transfection shows a faint band at a similar molecular weight and relatively same intensity with that of GN-CC. The authors should include blots with a better expression of Gr-CC and GN-CC constructs.

--- As suggested by the Reviewer, we performed a WB analysis of the lysates of cells expressing Gr-CC and GN-CC with an anti-GFP and another anti-collagen I body. The data obtained was shown as new Fig 1f.

4)The movie S2 is nice and provides the main result that only the processed collagen is secreted upon cleavage. However, a few mcherry spots are also seen to be secreted. The authors should solidify this result by performing a WB from the secreted medium and cell lysates and probe with GFP and mcherry antibodies, respectively.

--- As suggested by the Reviewer, we also detected a few secretion of mCherry (C-pp) in Movie S2, and explained it in the Discussion using the results of WB with media and cell lysates probed with an anti-GFP and mCherry antibody (p25, line 7-14). We also confirmed the secretion of the repeating structure domain (GFP) in comparisons of the signal intensities of WB as suggested (Fig 4g, see below).

Secretion assays should be ideally performed in a serum-free medium to avoid the BSA contamination, which affects WBs presented in Fig 4. As the authors further state that a small portion of Cpp gets processed and secreted, they should probe the same lane of cell lysates and supernatant with GFP and mcherry antibodies one after another and then express the data as a ratio of GFP/mcherry in cell lysates and supernatant. The expectation would be that this ratio would be much higher in the supernatant compared to the cell lysate.

--- We appreciate the suggestion by the Reviewer. We measured the signal intensities of Western blots with the lysates and culture media of cells expressing Gr-CC detected by an anti-GFP antibody. Since data obtained with an antibody (anti-GFP antibody) may be compared, we calculated the ratio of collagen/procollagen detected by the anti-GFP antibody, and confirmed the secretion of the repeating structure domain (Fig 4g). This result was described in the text (p15, line 10-14). The movement of mCherry (C-pp) was shown with measurements of the signal intensities of images and was described in the text (Fig. S8 and S10; page 13, line 10-13; page 14, line 13-16).

We attempted to use serum-free medium (medium with artificial serum) for WB with cell culture medium; however, background signals did not disappear.

5)The movie depicting the separation of Npp from Cpp is not very clear. Therefore, the authors should use the WB mentioned above strategy from cell lysates and secreted culture medium of cells expressing GN-CC construct and probing the blots using GFP and mcherry antibodies sequentially.

--- The fragment at 40 kDa, the size of which size corresponds to N-pp + GFP, may be detected in the cell lysate (Fig 4e); however, the signal is hard to see in the culture medium of cells expressing GN-CC because of the background signal of serum. We attempted to perform electrophoresis for a longer time with a larger volume of medium than in Fig 4f to separate tagged N-pp from the upper background band. We eventually detected the signal at approximately 40 kDa with the anti-GFP antibody, which confirmed that N-pp separates from C-pp (Fig S11) and also suggested that it is

secreted without degradation. The result obtained was described in the text (page 15, line 14-17; page 24, line 4-8). However, since signals including N-pp in WB with cell culture media were difficult to measure (e.g. Fig 4f), it was not appropriate to adopt the same method as in Fig 4g. To clearly show the separation of N-pp from C-pp, we added an enlarged merged image in new Fig 4d. We also measured the signal intensities of GFP and mCherry in the perinuclear region and each pseudopodium, and summarized the results in a graph in Fig S10. The result clearly showed the separation of N-pp from C-pp, and this was explained in the text (page 13, line 10-13)

6) Please add a better description of the relationship between TGF β and collagen processing and secretion. Currently, it appears out of the blue in the narrative. Similarly, explain AA2G in the text, and please expand the abbreviation before the first usage.

--- We apologize for these points. We revised the text with an explanation of TGF- β and AA2G (page 17, line 5- 11; page 17, line 11- line 14).

7) The way the data is presented, I cannot see any differences in cells in Fig 6D or the supplementary movies s7 and s8 with and without TGF β . Please show higher magnification images, and please use arrows to improve the presentation to point out what differences one expected to see.

--- We revised the data of shown in Fig 6d with a higher magnification of each image with or without TGF- β and added arrows to show the different color patterns in pseudopodia.

8) It must be re-emphasized that extensive edits in the text are necessary and should be scrutinized before resubmission. Some suggestions are as follows:

I would recommend to avoid expressions as on Page 19/ line 15: "region was yellow" and replace to "merged images of GFP and mCherry signal from cells expressing Gr-CC revealed high levels of colocalizations (that appears yellow on Video 1) which testifies that..." and Page 19/ line 21: "The green particles moved uni-directionally from the yellow perinuclear region to the tip of the pseudopodium..." could be changed to "vesicles containing processed procollagen with GFP tag (repeating domain structure) have been seen mostly in the periphery of the cell and were appearing in the perinuclear region and directing towards pseudopodium ". Page 23/ line 8. Page 35/line 4 and so on.

--- We are deeply grateful for the Reviewer's suggestions and revised the descriptions of the representations embodied above.

page 9, line 7-8; page 12 line 4 - 7; page 12 line 10-14; page 14, line 7-10; page 19, line 10-16; page 23, line 10-12.

9)Scale bars are missing. Figure 1C. 1D. 6B. 6C. 6D.

--- In accordance with the comment by the Reviewer, all scale bars were added.

Reviewer #2 (Comments to the Authors (Required)):

Collagens are necessary for the assembly of extracellular matrix and compose about 25% of our dry protein weight. The mechanism of secretion of these bulky cargoes is only recently beginning to emerge and the tagged collagens presented in this paper could be of tremendous value to the field. Protein TANGO1 has emerged as a key player in the collagen export pathway by its binding to collagens via Hsp47 and creating a novel route for their export.

This is an interesting paper and the authors are to be credited for address this important issue. The authors are encouraged to address the following issues to strengthen their conclusions.

1. They state that the newly formed trimer is packaged by the COPII cage with mono-ubiquitylated Sec31, and then transferred into the Golgi apparatus (6), where it undergoes final modification of its oligosaccharides.

The authors are encouraged to read up the literature on collagen export from the ER for accuracy and for a fair representation of the field.

--- In accordance with the comment by the Reviewer, we revised the description in the Introduction on the pathway of collagen transportation from the ER, in which TANGO1, Sec12, and Sec16 play important roles in the assembly of bulky COPII cargo (page 5, line 6-15).

2. Figure 1b. What are the green dots ? They appear to be ER exit sites? Show co-localization with an ER site marker. Use TANGO1 or Sec16.

--- In accordance with the comment by the Reviewer, we examined the colocalization of the tagged collagen (the repeating structure domain) with an ER marker (calreticulin), markers for the ER exit site (Sec16a) and the cis-Golgi (GM130) to elucidate the pathway. In the revised manuscript, we performed a collagen trafficking assay in which cells were treated at 40°C for 3 hours for the accumulation of cargo, including tagged collagen, in the ER, followed by 32°C for 16 hours for transport into the Golgi. These results showed that GFP-tagged collagen initially localized in the ER at 40°C, and was then transported to the Golgi at 32°C via ER exit sites, as expected by the Reviewer. The results of immunofluorescent experiments using antibodies against calreticulin, Sec16a and GM130 (another image) are shown as Fig 1d and Fig S2.

3. Figure 1d. Very little co-localization of expressed collagens with GM130. The rest and the predominant material is probably still in the ER or ER exit sites. Please check. I would suggest the authors generate stable lines to avoid issues due to non-uniform expression of collagens.

--- We performed the collagen trafficking assay with NIH3T3 cells stably expressing Gr-CC. The tagged collagen initially localized in the ER at 40°C (Fig 1d) and was transported to the Golgi at 32°C (Fig 1e) via ER exit sites (Fig S2). Therefore, the GFP signals (the repeating structure domain) out of the cis-Golgi in previous Fig. 1d (new Fig S3) appear to exist in the ER or the ER exit site, as suggested by the Reviewer.

4. Figure 3B. What is the arrow suppose to show? I do not see anything in the images shown. Is this green spot a transport carrier? Please provide better pictures at higher magnification. Provide some sort of quantitation? Are the presumed spots carriers between ER and the Golgi or post Golgi?

--- We revised the images with the higher magnification of a particle containing processed collagen with GFP (the repeating structure domain). We also attempted to perform live imaging using an objective lens with a higher magnification (64×, 100×); however, we lost track of the particles containing GFP because of the shallow depth of the field. We generally conduct live imaging with 20× objective lens (Plan APOCHROMAT 20×/0.8, Zeiss) with large frame sizes (2048 × 2048 dpi), and magnify the points being observed. This was described in the Materials and Methods (page 30, line 1-4).

The particle may be traced in pseudopodium (left and middle panel of Fig 3b), and is secreted out of the cell (right panel of Fig 3b). Therefore, granules containing the GFP signals in pseudopodia were considered to be carriers after the Golgi.

5. Where is the N-pp degraded after cleavage?

--- In the original manuscript, we stated that N-pp is degraded during transportation in the cell and/or after its secretion, but need to change this result because the specific signal approximately 40 kDa was eventually detected in a Western blotting analysis of the culture medium of cells expressing GN-CC (Fig S11). Therefore, N-pp was not degraded after cleavage. This result was described in the text (page 15, line 14-17; page 24, line 4-6).

6. the authors should show whether the cleavage takes places in the ER or the Golgi. use of specific means to inhibit collagen egress from these compartments will help address this important concern.

--- To identify the organelle at which the intracellular cleavage of procollagen occurs, we performed an experiment in which collagen trafficking was inhibited at the ER in cells treated with bafilomycin (Fig 3c). The result obtained revealed the presence of GFP-collagen is in the ER, but not mCherry-C-pp, indicating that the C-pp is cleaved in the ER. This result was explained in the text (page 12, line 14-18).

Reviewer #3 (Comments to the Authors (Required)):

Tanaka et al. describe the cloning and characterisation of a human COL1A1 cDNA construct that has GFP integrated in the repeating structure domain. This work attempts to overcome the problem that N- or C- terminal tagging of COL1 does not allow monitoring the deposition of collagen in the ECM, as both, N- and C- terminus of COL1 are cleaved by proteases for proper processing of procollagen into collagen fibers.

Tagging COL1 in its repeated structure domain has remained a challenge, mainly due to concerns of proper folding and hindrance in fibril formation, which the authors of the current manuscript have summarised very well.

In the current manuscript, light as well as electron microscopy techniques have been used to analyse the cellular localisation of Gr-CC encoded COL1A1 in NIH3T3 cells. Western blot analysis was used to test the folding of Gr-CC encoded COL1A1 in cell lysates followed by measurements of GFP levels in cells, supernatants as well as ECM. An alternative construct driving the COL1A1 expression by a CMV promotor to one which should respond to TGF β 1 stimulation is also described.

If the current manuscript could convincingly show that the COL1A1 with a GFP tag in the repeating structure domain can fold properly and assemble into fibrils in the ECM, like wildtype COL1A1, it would represent an important major methodological advancement opening a number of new experimental approaches with relevance to e.g. ECM formation and regulation or development of fibrosis.

While the methods of choice to validate the claims in the current work appear suitable, a substantial fraction of the data shown are in this reviewer's view not of sufficient quality to substantiate the claims made by the authors. The following points need to be seriously addressed before this manuscript should be published:

1) Authors should publish the sequence of the insertion of the tag, this should subsequently help the community to endogenously tag COL1A1 repeating structure domain. Authors should mention why this site was chosen, was it a random choice or there was some methodology involved.

--- We attempted more than ten constructs, each of which had EGFP and mCherry inserted into different points of human type I pre-procollagen α 1, and eventually selected the constructs in Fig. 1a using the experiments described in the manuscript. The inserted points were semi-randomly selected. We cannot publish the inserted sequences of EGFP and mCherry in human type I pre-procollagen α 1 because the constructs are valuable for a wide range of industries, such as

pharmaceutical and cosmetic agents, and we do not have a patent for them in countries other than Japan. We explained how we selected the constructs in the text (page 8 line 13-17; page 29, line 6-9).

2) Figure 1C: A control EM micrograph showing cells mock transfected is missing. In the micrographs shown, the ER appears bloated, which suggests a transport block of the GFP tagged COL1, possibly caused by the tagging? and without a control reference, it is not possible to judge the validity of the experiment.

--- Regarding a control reference for immuno-EM in Fig 1c, we simultaneously performed a control EM without the primary antibody, and added the image obtained to revised Fig 1c.

In accordance with comment by the Reviewer, to establish whether a transport block of GFP-tagged COL1 occurs, we examined the expression of the ER stress marker CHOP with a Western blot analysis. The data obtained showed that the expression of CHOP was not upregulated in the absence of tunicamycin (Fig S4), supporting proper protein folding in the NIH3T3 cells used in Fig 1c. We used NIH3T3 cells stably expressing Gr-CC for immuno-EM, which grow normally, reflecting no transport block in the ER. To compare the ER between parental NIH3T3 and its cell line stably expressing Gr-CC, we performed ER staining with CellLight kit (Thermo fisher, Fig S1). The data obtained showed no marked differences. We checked reference papers showing the proper condition of the ER in NIH3T3 cells: reference EM images (Fig 5c in J Biomed Mater Res Part A 2016:104A:272–282, Fig 5A in PLoS ONE 13(5): e0196649 (2018)) indicate the ER in proper NIH3T3 cells without transfection up to 500 nm. These data showed that the ER in Fig 1c is not abnormal. We explained these points in the revised text (page 9, line 9-11; page 9, line 11- page 10 line 5; page 21, line 10-page 22, line 1).

3) Figure 1D and related supplementary figure: the colocalization of Gr-CC encoded COL1A1 with GM130 is not convincing and presented images should be improved and accompanied with a quantitative evaluation of the co-localisation, e.g. by plot profiles of the two signals (GM130 and Gr-CC GFP).

--- To obtain clear images, we performed the collagen trafficking assay in which cells treated at 40°C for 3 hours showed the arrested transportation of synthesized proteins at the ER, and their continuous treatment at 32°C released this arrest, resulting in accumulation of proteins in the Golgi. Revised immunofluorescence images with an anti-GM130 antibody, in which the colocalization of collagen containing the GFP tag with GM130 was more clearly shown than in the previous image, were added as Figs 1e and S2.

4) In all the images throughout the manuscript, Gr-CC, the GFP and mCherry colocalise remarkably

but not in figure 1e. Only one cell is shown and corresponding signals do not overlap. Again, colocalization quantification should be done.

--- We apologize for presenting an inadequate image as previous Fig 1e, in which C-pp signals were not clear because of the low resolution of the image. The revised image shown as Fig S9 confirmed that GFP and mCherry colocalized in the perinuclear region.

In accordance with the comment of the Reviewer, we measured the signal intensities of mCherry and GFP in Fig 4a, showing the colocalization of GFP and mCherry in the perinuclear region, and confirmed predominant accumulation of mCherry (C-pp) in the perinuclear region and predominant secretion of GFP (the repeating structure domain) via pseudopodia (Fig S8).

Also, co-localisation of Gr-CC and lysotracker could be done to exclude Gr-CC localising to lysosomes. Alternatively, lysosomal inhibitors such as chloroquine or bafilomycin should be used. Furthermore, levels of ER stress markers such as BiP and CHOP could be measured to ensure that indeed proper folding has occurred and that no unfolded protein response has been triggered.

--- We stained lysosomes with LysoBrite dye (Cell Navigator Lysosome staining kit), which is more photostable than Lyso Tracker dyes (Fig S9). The signal intensities of GFP and mCherry in lysosomes were quantitated, and we detected more mCherry (C-pp) in the lysosomes than GFP (the repeating structure domain) (21.1 and 5.94% of each signal in the whole cell, respectively), indicating the selective incorporation of cleaved C-pp into lysosomes after the proper processing of the tagged procollagen. These data have been described in the caption of Fig S9.

In accordance with the comment by the Reviewer, the ER stress marker CHOP was examined using a Western blot analysis, and the results obtained showed that the expression of CHOP was not upregulated in the absence of tunicamycin (Fig S4). These results supported proper protein folding in NIH3T3 cells with the tagged procollagen.

5) Figure 1f: This reviewer is not convinced that there is a band in the Gr-CC lane upon hCol1 immuno-blot. Please show where the endogenous collagen runs, using anti-mouse collagen antibody for a reference and estimation of expression levels.

--- According to the comment by the Reviewers, we revised the Western blotting image with the anti-human collagen 1 α 1 antibody in Fig 1f.

To obtain better images for the Western blot analysis with the anti-human collagen 1 α 1 antibody, we checked more than five commercial antibodies for collagen 1 α 1; however, most showed many non-specific signals in the Western blot analysis and, thus, were not be used. We eventually selected an antibody supplied by Boster Biological Technology, which exhibits reactivities with the

collagen 1 α 1 proteins of humans, mice, and rats, and revised the figure (new Fig 1f). However, since the amino acids of the antigen domain have several differences between humans and mice, the signal intensity of endogenous collagen 1 α 1 in Fig 1f may result in the underestimation of its quantity in mouse NIH3T3 cells. We explained this point in the text (page 10, line 7-9).

6) Figure 2 and Figure S2: to convincingly show that Gr-CC is deposited in the ECM, the authors should use macromolecular crowding protocol (<https://doi.org/10.3389/fmed.2020.615774>).

--- We appreciate the Reviewer for providing this information, and used PVP40 to gain clearer images of fibrils containing GFP-tagged collagen out of the cells. We have gained better images and represented one in the revised Supplementary materials (Fig S5).

7) The movies and data showing the movement of different signals are not optimal. Most of the images are saturated. Removing the transmitted light channel and just showing individual color channels with a black background could improve the visualisation.

--- Since the signal intensity in the perinuclear region was markedly stronger than that in the pseudopodia, it was difficult to show signals in the pseudopodia without the overexposure of signals in the perinuclear region. Signal intensities of Figs 3a, 4a, 4d, 6b and 6d (and related movies to these figures) were adjusted to show the colors of the particles in pseudopodia; therefore, signals in the perinuclear region were slightly saturated. Image cropping around pseudopodia may be used to resolve this issue in these figures; however, since we would like to show the color alteration from yellow in the perinuclear region to green in pseudopodia, we allowed some overexposure of signals in the perinuclear region. We need to adjust the whole signal intensity in accordance with the portion to be shown with the figure, and attempted to avoid saturated images as much as possible. To show an unsaturated image of the colocalization of GFP with mCherry in the perinuclear region as a reference, we prepared a new image in Fig 1b, in which we adjusted the signal intensity to the merged signals in yellow in the perinuclear region, which resulted in faint signals of green particles in pseudopodia (new Fig 1b, indicated by arrows). As a reference, images of hepatic stellate cells, in which GFP and mCherry signals were not saturated in the perinuclear region, are shown in revised Fig S18.

8) Figure 4a and Movie S5: both, GFP and mCherry show the same movement. A difference between N-pp and C-pp is not clear to this reviewer. Based on the data shown, the claim that N-pp and C-pp have different fates is not valid.

--- (The distribution of N-pp and C-pp is addressed in Fig 4d and Movie S4.) We quantified the signal intensities of GFP and mCherry in pseudopodia and the perinuclear region of a representative cell in Fig 4d, and this result was summarized in graphs (Fig S10). The data obtained showed that

more GFP (N-pp) signals exist than mCherry (C-pp) signals in pseudopodia. This result was explained in the text (page 14, line 13-16).

9) Figure 4C: loading controls of the western blot cell lysates are missing.

--- Regarding the controls required for Fig 4c, we used the medium of mock-transfected cells and medium without cells as negative controls, and the purified mCherry protein as a positive control. We also used the purified GFP to confirm the specificity of the anti-mCherry antibody. The results of the Western blot analysis with the lysates of cells expressing Gr-CC and GN-CC are shown as Fig 4b with the lysate of mock-transfected cells and the mCherry protein as negative and positive controls, respectively. Since the Western blot analysis with the lysates of cells expressing Gr-CC and GN-CC may be overlapping data that increase the complexity of the image in Fig 4c, we do not want to reuse these cell lysates for Fig. 4c. We assume that the Reviewer expects an identical signal size at approximately 25 kDa in Fig 4b and 4c. However, the signal at approximately 25 kDa in Fig 4b may not be identical to that in Fig 4c. A secreted protein does not accumulate in the cells, resulting in a very small quantity in the cells to be detected by a Western blot analysis.

10) Exchanging the normal CMV promotor for one which responds to TGFb is a very nice idea. Here again, the authors should submit the sequences they have used. This construct can help also accelerate the deposition of Gr-CC in macromolecular crowding experiment.

--- We cannot publish the inserted sequences of EGFP and mCherry in human type I pre-procollagen $\alpha 1$ for the reasons described above.

According to the Reviewer, we performed the macromolecular crowding experiment using cells expressing col1p_Gr-CC/NIH3T3, and represented the result in Fig S5.

11) Figure 5C: data shown suggest that there is no difference in GFP levels in control and TGF β 1 treated samples and suggests to this reviewer that the data do not support the hypothesis that Gr-CC is secreted into the extracellular space. The existence of GFP in the culture supernatant (figure 5a and 5b) could also be explained to occur due to lysis of GFP positive cells over time in culture. The increase in the intracellular level in figure 5d is not reflected in Figure 5C. Therefore, based on these data, the claim that collagen has differential kinetics under disease conditons such as in hepatic fibrosis cannot be made. This data suggest more that the Gr-CC is not secreted.

--- In the revised manuscript, we added data to Fig 4g to show that the collagen/procollagen ratio markedly differed between the cell lysate and cell culture medium. These data demonstrated that the tagged procollagen was processed in the cells and the tagged collagen was secreted into the extracellular space. The data shown in Fig 5b are consistent with those in Fig 4g. We added this information in the text (page 15, line 10-14). Therefore, the graph in Fig. 5c indicates

time-dependent increases in collagen levels containing the GFP tag in control and TGF- β 1-treated samples with three independent experiments (none, 2.0 ng/ml TGF- β 1 and 12.5 ng/ml TGF- β 1), which indicated that processed collagen containing the GFP tag was secreted into the extracellular space.

Since cells were not in over-growth conditions, as shown in Fig 5d, and few cells died, the simultaneous increases observed in GFP levels in the three cell culture media in Fig 5c do not appear to be caused by cell lysis. TGF- β 1 has been shown to upregulate the expression of the promoter of human type I collagen α 1 (JBC 269, 12684-91, 1994); therefore, the increase observed in GFP levels in cells expressing the Gr-CC may be predicted. However, the TGF- β 1 treatment of cells did not increase GFP levels in the cell culture medium, as shown in Fig 5c, indicating that TGF- β 1 had no effect on the secretion of collagen. These explanations were added to the revised manuscript (page 17, line 4 - 16).

12) Images showing the expression of Gr-CC in hepatic stellate cells are saturated.

--- In accordance with the comment by the Reviewer, we obtained images of hepatic stellate cells expressing Gr-CC, the signal intensities of which were adjusted to those in the perinuclear region, as shown in Fig S18.

We also revised the signal intensities of the images in Fig 6 as much as possible, and added enlarged panels to clearly show the color in the pseudopodia (Fig 6d). We adjusted the whole signal intensity in accordance with the portion to be shown as described above (#7). Since the signal intensities of Fig 6b and 6d were adjusted to show the color of particles in pseudopodia, signals around the perinuclear region became saturated.

December 23, 2021

Re: Life Science Alliance manuscript #LSA-2021-01060R

Dr. Toshiaki Tanaka
Tokyo Institute of Technology
School of Life Science and Technology
4259 Nagatsuta
Midori-ku
Yokohama, Kanagawa 226-8501
Japan

Dear Dr. Tanaka,

Thank you for submitting your revised manuscript entitled "Visualized procollagen Ia1 demonstrates the intracellular processing of propeptides" to Life Science Alliance. The manuscript has been seen by the original Reviewers whose comments are appended below. While the Reviewers continue to be overall positive about the work in terms of its suitability for Life Science Alliance, some important issues remain.

Our general policy is that papers are considered through only one revision cycle; however, given that the suggested changes are relatively minor, we are open to one additional short round of revision. Please note that I will expect to make a final decision without additional reviewer input upon resubmission.

Please submit the final revision within one month, along with a letter that includes a point by point response to the remaining reviewer comments.

To upload the revised version of your manuscript, please log in to your account: <https://lsa.msubmit.net/cgi-bin/main.plex>
You will be guided to complete the submission of your revised manuscript and to fill in all necessary information.

B. MANUSCRIPT ORGANIZATION AND FORMATTING:

Sincerely,

Reviewer #1 (Comments to the Authors (Required)):

I greatly appreciate the time and effort that the authors put to address the questions and comments. Despite most of the question were addressed there are some points that require some attention

1) Because current study emphasizes the maturation and processing that is associated with subsequent cleavage of maturing chains, it would be nice to show cleavage sites on protein diagram on Figure 1A.

2) Page 30/ line 1 and 2 indicates that "GR CC* used in 15 Fig. 4e lacks the cleavage sequence for C-pp processing". The question: is that statement true for all experiments with the mentioned construct? If so, please indicate that statement in legend for figure 1A and put a mark in figure.

3) Movie 1. I am wondering about mCherry positive events that are seen in the video, that creates the impression that c-pp or mCherry itself probably undergo secretion.

4) Figures 3A and 4A seem to be duplicating information. Please specify the difference. (In both cases same markers were used).

5) Please provide more convincing co-localization data within Golgi. Probably perform a 20C block to accumulate transgenes in TGN with staining against TGN46 (colocalization data).

6) Suppl. Figure S3. Please use arrows to highlight differences.

Reviewer #2 (Comments to the Authors (Required)):

I thank the authors for their efforts in addressing many of my concerns. However, there are two issues that remain unaddressed.

1. In figure 1 e, the authors propose that collagen is localised to GM130 containing structures. This is not convincing. The magenta particles (GM130) are in the same region , but not co localized to the green (collagen) elements. Also, vast majority of the collagen is still in the ER. The authors should treat cells with cycloheximide to prevent new synthesis and to flush the ER pool into the next secretory compartment.

2. Bafilomycin treatment usually affects cargo egress from the late Golgi and entry/degradation of proteins in the endolysosomal compartments. I am surprised to see the effect of bafilomycin at the level of the ER. The authors should test this more rigorously.

Response to Reviewers' Comments

Reviewer #1 (Comments to the Authors (Required)):

I greatly appreciate the time and effort that the authors put to address the questions and comments. Despite most of the question were addressed there are some points that require some attention

1) Because current study emphasizes the maturation and processing that is associated with subsequent cleavage of maturing chains, it would be nice to show cleavage sites on protein diagram on Figure 1A.

--- We indicated the cleavage sites in revised Fig 1a with white triangles.

2) Page 29/ line 15 and 16 indicates that "GR CC* used in 15 Fig. 4e lacks the cleavage sequence for C-pp processing". The question: is that statement true for all experiments with the mentioned construct?

If so, please indicate that statement in legend for figure 1A and put a mark in figure.

--- Because Gr-CC* was only used in Fig 4e and not used in other experiments in the present manuscript, we think that the construct is not appropriate to be described in Fig 1a. We thus described the explanation of Gr-CC* in legend for Fig 4e and removed it from Materials and Methods (p29, line 14).

3) Movie 1. I am wondering about mCherry positive events that are seen in the video, that creates the impression that c-pp or mCherry itself probably undergo secretion.

--- In our previous response addressing Reviewer #1's comment 4, we have explained a few secretion of mCherry (C-pp) in the Discussion with data in Fig 4a, Movie S3, and the results of WB with cell lysates and media probed with an anti-mCherry antibody in Figs 4b and c (p25, line 12-18).

4) Figures 3A and 4A seem to be duplicating information. Please specify the difference. (In both cases same markers were used).

--- To show high levels of colocalization of GFP and mCherry signals from cells expressing Gr-CC, and to reveal intercellular processing of procollagen at the perinuclear region (p12, line 9-15), the individual images of GFP and mCherry were not essential in Fig 3a. We therefore removed individual images of GFP and mCherry signals in Fig 3a so that duplicating information indicated by the Reviewer has been solved.

5) Please provide more convincing co-localization data within Golgi. Probably perform a 20C block to accumulate transgenes in TGN with staining against TGN46 (colocalization data).

--- In accordance with the suggestion by the Reviewer, we performed another collagen trafficking assay at 20°C with staining against anti-TGN46 antibody to show clearer colocalization of tagged collagen within Golgi. We added the data as new Fig 1e and explained the results in the text (p10, line 2-5; p21, line 15- p22, line 1).

6) Suppl. Figure S3. Please use arrows to highlight differences.

--- We removed previous Fig S3; instead, we added another data showing colocalization of the repeating structure domain (GFP) with GM130. Colocalization is highlighted with arrows in new Fig S3.

Reviewer #2 (Comments to the Authors (Required)):

I thank the authors for their efforts in addressing many of my concerns. However, there are two issues that remain unaddressed.

1. In figure 1 e, the authors propose that collagen is localised to GM130 containing structures. This is not convincing. The magenta particles (GM130) are in the same region, but not co localized to the green (collagen) elements. Also, vast majority of the collagen is still in the ER. The authors should treat cells with cycloheximide to prevent new synthesis and to flush the ER pool into the next secretory compartment.

--- We performed an immunostaining analysis with anti-TGN46 antibody in the presence of cycloheximide instead of GM130, to show clearer colocalization of tagged collagen with Golgi (trans-Golgi). We added the data as new Fig 1e and explained the result in the text (p10, line 2-5).

2. Bafilomycin treatment usually affects cargo egress from the late Golgi and entry/degradation of proteins in the endolysosomal compartments. I am surprised to see the effect of bafilomycin at the level of the ER. The authors should test this more rigorously.

--- We are deeply sorry for describing erroneous information. Bafilomycin was NOT used in the present experiments. We used brefeldin A to arrests protein trafficking at the ER. We revised the erroneous description in the manuscript. We are deeply grateful for the Reviewer to find the mistake.

January 20, 2022

RE: Life Science Alliance Manuscript #LSA-2021-01060RR

Dr. Toshiaki Tanaka
Tokyo Institute of Technology
School of Life Science and Technology
4259 Nagatsuta
Midori-ku
Yokohama, Kanagawa 226-8501
Japan

Dear Dr. Tanaka,

Thank you for submitting your revised manuscript entitled "Visualized procollagen Ia1 demonstrates the intracellular processing of propeptides". We would be happy to publish your paper in Life Science Alliance pending final revisions necessary to meet our formatting guidelines.

- please upload your main and supplementary figures as single files, one file per figure. They should not consist of more parts.
- all figure files should be uploaded as individual ones, including the supplementary figure files and all figure legends should only appear in the main manuscript file
- please use the [10 author names, et al.] format in your references (i.e. limit the author names to the first 10)
- please use capital letters when introducing panels in figures, their legends, and callouts in the manuscript text
- on page 24, you mention that bafilomycin was used in Figure 3C. I believe this should say brefedlin A

FIGURE CHECKS:

-In Figure 1C, it looks like the box shown on the image with the 5um bar could be positioned slightly better to reflect the magnified image shown with the 2um bar.

A. FINAL FILES:

B. MANUSCRIPT ORGANIZATION AND FORMATTING:

Sincerely,

January 26, 2022

RE: Life Science Alliance Manuscript #LSA-2021-01060RRR

Dr. Toshiaki Tanaka
Tokyo Institute of Technology
School of Life Science and Technology
4259 Nagatsuta
Midori-ku
Yokohama, Kanagawa 226-8501
Japan

Dear Dr. Tanaka,

Thank you for submitting your Research Article entitled "Visualized procollagen Ia1 demonstrates the intracellular processing of propeptides". It is a pleasure to let you know that your manuscript is now accepted for publication in Life Science Alliance. Congratulations on this interesting work.

DISTRIBUTION OF MATERIALS:

Again, congratulations on a very nice paper. I hope you found the review process to be constructive and are pleased with how the manuscript was handled editorially. We look forward to future exciting submissions from your lab.

Sincerely,
